# Derivative Informed Learning of Exchange-Correlation Functionals

Eike S. Eberhard [1 2 3]   Luca A. Thiede [4 5]   Abdul Aldossary [4]   Andreas Burger [4 5]   Nicholas Gao [6]
Vignesh Bhethanabotla [7]   Alán Aspuru-Guzik [4 5]   Stephan Günnemann [1 2 3]

## Abstract

Machine-learned (ML) XC functionals aim to replace human-designed density functional approximations by learning directly from reference data, but they still do not consistently outperform traditional $\mathcal{O}(N^4)$-scaling hybrid functionals. We therefore study a hybrid-distillation setting, where $\mathcal{O}(N^3)$-scaling semilocal ML-XC functionals are trained to reproduce B3LYP/def2-SVP targets. We introduce *Derivative Informed XC-Loss* (DI-Loss), a loss that incorporates additional information from the reference hybrid functional by supervising first and second derivatives of the energy on the Grassmannian of admissible density matrices. Rather than only matching the self-consistent fixed point, DI-Loss aligns the local first- and second-order response of the learned functional with that of the target functional. Across four evaluated architectures, DI-Loss consistently improves the main energy metrics. Averaged uniformly across architectures, the total-energy MAE decreases by 66% relative to energy and density supervision alone. The density-sensitive mean-field energy metric $E_\rho$ improves from 1.2 to 0.8 mEh on average, while dipole and $\mathcal{L}_2$ density errors do not improve uniformly. We further show that densities from the distilled functionals reduce hybrid-functional SCF iterations by up to 55%. In downstream TDDFT calculations, Hessian supervision improves excited-state predictions, with XCdiff reducing the mean excitation-energy MAE by 24–35% across molecule sizes on QM40.

[1]Technical University of Munich, Munich, Germany [2]Munich Data Science Institute, Munich, Germany [3]Munich Center for Machine Learning, Munich, Germany [4]University of Toronto, Toronto, Canada [5]Vector Institute, Toronto, Canada [6]CuspAI, Berlin, Germany [7]California Institute of Technology, Pasadena, United States of America. Correspondence to: Eike S. Eberhard <eike.eberhard@tum.de>.

*Proceedings of the 43$^{rd}$ International Conference on Machine Learning*, Seoul, South Korea. PMLR 306, 2026. Copyright 2026 by the author(s).

## 1. Introduction

The ability to accurately and efficiently predict molecular and materials properties is central to modern computational chemistry. Such predictions enable the rational design of molecules and materials across a wide range of applications, from catalyst design and energy storage to drug discovery (Sliwoski et al., 2014; Lu, 2021; Aldossary et al., 2024; Rowaiye et al., 2025). A fundamental challenge in this domain is the tension between predictive accuracy and computational scalability. In principle, most of chemistry is described by the Schrödinger equation which admits an exact solution (Knowles & Handy, 1984). However, its factorial scaling with system size renders it infeasible beyond very small systems. Highly accurate approximations such as coupled-cluster theory (Purvis III & Bartlett, 1982; Raghavachari et al., 1989) reduce this cost, but still scale steeply with system size, typically as $\mathcal{O}(N^6)$ for CCSD and $\mathcal{O}(N^7)$ for CCSD(T). Density functional theory (DFT) occupies a central position in electronic structure theory by offering a favorable compromise between accuracy and cost.

Density Functional Theory (DFT) is based on the Hohenberg-Kohn (HK) theorems (Hohenberg & Kohn, 1964), which reduce the description of an interacting many-electron system from its $3N$-dimensional wavefunction to the three-dimensional electron density $\rho(\mathbf{r}) : \mathbb{R}^3 \to [0, \infty)$. The first theorem establishes a one-to-one correspondence between the external potential $v_{\text{ext}}(\mathbf{r})$ generated by the nuclei and the ground-state density $\rho_0(\mathbf{r})$. The second theorem guarantees the existence of an energy functional $E[\rho]$ whose minimization yields the ground-state density as the minimizer and the ground-state energy as the minimum value. The only system-specific input is the nuclear potential $v_{\text{ext}}(\mathbf{r})$, while the remaining contributions form a universal functional shared by all electronic systems.

While the HK theorems guarantee the existence of $E[\rho]$, they say nothing about its form, leaving its parameterization as the central challenge of DFT. Kohn and Sham (Kohn & Sham, 1965) proposed mapping the interacting system onto a fictitious non-interacting reference system with the same ground-state density $\rho^*$, whose kinetic energy $T_s$ can be evaluated exactly through *single-particle orbitals* $\varphi_i$ :

$\mathbb{R}^3 \to \mathbb{R}$. In this setting, the total energy functional can be decomposed as

$$E[\rho] = E_{\text{nuc}} + T_s[\rho] + E_{\text{C}}[\rho] + E_{\text{xc}}[\rho] \,, \qquad (1)$$

where the nuclear repulsion energy $E_{\text{nuc}}$ and the Coulomb (Hartree) energy $E_{\text{C}}$ are known exactly, while all many-body effects beyond classical electrostatics, together with the residual kinetic contribution $T[\rho] - T_s[\rho]$, are absorbed into the unknown exchange-correlation (XC) functional $E_{\text{xc}}$. The density itself is reconstructed from the orbitals via $\rho(\mathbf{r}) = \sum_i |\varphi_i(\mathbf{r})|^2$, with the orbitals obtained by solving the Kohn-Sham equations

$$\{h_{\text{core}}(\mathbf{r}) + v_C[\rho](\mathbf{r}) + v_{\text{xc}}[\rho](\mathbf{r})\} \varphi_i = \varepsilon_i \varphi_i \,, \quad (2)$$

subject to the orthonormality constraint $\int \varphi_i(\mathbf{r}) \varphi_j(\mathbf{r}) \, d\mathbf{r} = \delta_{ij}$. KS-DFT retains a favorable computational scaling, making it the workhorse of modern electronic structure calculations.

The accuracy of KS-DFT is almost solely limited by the quality of the approximation to the unknown XC functional

$$E_{\text{xc}} : \left\{ \rho : \mathbb{R}^3 \to \mathbb{R}^+ \,\middle|\, \int \rho(\mathbf{r}) \, d\mathbf{r} = N_{\text{elec}} \right\} \to \mathbb{R} \,, \quad (3)$$

which maps the electron density to a scalar energy contribution. Recent efforts have attempted to bypass traditional, human-designed density functional approximations (DFAs) by learning XC functionals directly from high-accuracy reference data (Snyder et al., 2012; Nagai et al., 2020; Dick & Fernandez-Serra, 2021; Gao et al., 2024), training the functional to reproduce ground-state densities and energies obtained from reference calculations such as coupled-cluster. While promising, these ML-driven functionals generally compete with, but do not consistently surpass, traditional $\mathcal{O}(N^4)$-scaling *hybrid* functionals on energy benchmarks (Luise et al., 2025; Karton, 2026).

In this work, we target the accuracy-efficiency gap by *distilling* a traditional $\mathcal{O}(N^4)$-scaling XC functional into a lower-cost $\mathcal{O}(N^3)$-scaling ML XC functional. The advantage of this approach is the access to additional information about the reference functional in the vicinity of the ground state, which is not available from expensive *ab initio* reference calculations. Rather than supervising only energies and densities as in prior work, we additionally supervise the first and second functional derivatives of the energy with respect to the density, aligning the self-consistent field (SCF) dynamics of the distilled functional with those of the target. These additional loss terms enable the distillate to more faithfully reproduce ground-state energies and improve out-of-distribution generalization, as well as response properties relevant to excited-state calculations. While we restrict ourselves to distilling $\mathcal{O}(N^4)$-scaling functionals in this work, the approach extends to more accurate functionals such as double hybrids (Grimme, 2006).

## 2. Background

**The discretization of KS-DFT** utilizes a finite basis $\{\chi_\mu : \mathbb{R}^3 \to \mathbb{R}\}_{\mu=1}^B$ to solve Eq. (2) in a $B$-dimensional space (Lehtola et al., 2020). In this discretization, the set of eigenfunctions $\boldsymbol{\varphi}(\boldsymbol{r}) = (\varphi_1, ..., \varphi_B)^T$ can be represented using an *orbital coefficient matrix*

$$\boldsymbol{\varphi}(\boldsymbol{r}) = C^T \boldsymbol{\chi}(\boldsymbol{r}) \,. \qquad (4)$$

Notably, only the first $O = N_{\text{elec}}/2$ orbitals are *occupied* by electrons, the remaining $V = B - O$ are commonly referred to as *virtual*.[1] We denote the common slices $\boldsymbol{C}_{:,:O}$ and $\boldsymbol{C}_{:,O:}$ as $\boldsymbol{C}_{\text{occ}}$ and $\boldsymbol{C}_{\text{virt}}$, respectively. The electron density can then be written as $\rho = \boldsymbol{\chi} \boldsymbol{C}_{\text{occ}} \boldsymbol{C}_{\text{occ}}^T \boldsymbol{\chi}$, making it convenient to define the *density matrix* $\boldsymbol{P} = \boldsymbol{C}_{\text{occ}} \boldsymbol{C}_{\text{occ}}^T$.

Starting from some $\rho^{(t)}$, one can plug it back into Equation (2) to obtain a new set of orbitals by filling the lowest-energy orbitals up to $N_{\text{elec}}/2$, constructing a new $C_{\text{occ}}^{(t+1)}$ corresponding to $\rho^{(t+1)}$. This procedure is repeated until self-consistency, i.e., until $\rho^{(t)} = \rho^{(t+1)}$, yielding the ground state density, $\rho_{\text{gs}}$.

From the way we defined $P$, it is clear that not any matrix in $\mathbb{R}^{B \times B}$ is a valid density matrix. Interestingly, the set of conditions implicitly defines a $O_{\text{basis}}/(O_{\text{occ}} \times O_{\text{virt}})$ dimensional manifold, where $O_{\text{basis}}$ is the orthogonal group of the size of the basis, also known as the Grassmannian manifold (Edelman et al., 1998). This manifold can be understood as the space of projection matrices of the occupied $O$, invariant to unitary rotations within $O$ or $V$. Starting from a valid coefficient matrix $C_0$ on this manifold, all other valid $C$ matrices can be parameterized by

$$C(\theta_{ov}) := C_0 \exp \begin{pmatrix} 0 & \theta_{\text{ov}} \\ -\theta_{\text{ov}}^T & 0 \end{pmatrix} \,, \qquad (5)$$

where $\theta_{\text{ov}} \in R^{O \times V}$ are the so-called *orbital rotation angles*. For more details and derivations, refer to Section A.

**Traditional XC-functional approximations** have historically been parameterized as $E_{\text{xc}} = \int \varepsilon_{\text{xc}}[\rho](\mathbf{r}) \, \rho(\mathbf{r}) \, d\mathbf{r}$, modeling an *intensive* energy density $\varepsilon_{\text{xc}}$ that depends on local properties of the electron density (Appendix E). Beyond the basic Local Density Approximation (LDA) solely based on $\rho$, Generalized Gradient Approximations (GGAs) add dependence on the density gradient $\nabla\rho$, while meta-GGAs further include the kinetic energy density $\tau$, all of which retain $\mathcal{O}(N^3)$ scaling. Higher accuracy can be achieved by incorporating exact exchange from Hartree-Fock theory, which is done for (range-separated) *hybrid* functionals (Becke, 1993; Yanai et al., 2004). *Double hybrids* (Grimme, 2006) add perturbative correlation corrections at even greater expense.

---

[1]Here we limit ourselves to restricted/closed-shell systems typical for stable organic chemistry, which have an even number of electrons.

This increased accuracy, however, comes at a high computational cost: Hybrid and range-separated hybrid functionals scale as $\mathcal{O}(N^4)$, while double hybrids scale as $\mathcal{O}(N^5)$. As a result, the most accurate KS-DFT-based methods sacrifice some of the favorable scaling that originally motivated it.

**Time-dependent DFT (TDDFT)** is the most widely used method for calculating excited-state properties of molecular systems. It is formally grounded in the Runge-Gross theorem (Runge & Gross, 1984), the time-dependent analogue of the Hohenberg-Kohn theorem, which establishes a one-to-one correspondence between the time-dependent external potential and the time-dependent electron density. Intuitively, a time-varying electric field, such as an oscillating light wave, perturbs the ground-state density and induces a density response. If the field frequency $\omega$ matches a natural transition frequency of the system, this response becomes singular (a "pole"), and the frequencies at which these poles occur correspond to the molecule's excitation energies. In linear-response TDDFT, computing these poles reduces to a generalized eigenvalue problem whose central operator is the orbital Hessian (Gross & Maitra, 2012). Its XC contribution is precisely the second functional derivative of $E_{\mathrm{xc}}$ with respect to the density. TDDFT excitations can therefore probe how well an ML functional reproduces the curvature of the reference functional around the ground-state density. Computational details are provided in Section F.

## 3. Related Work

**Data-driven DFAs** predate recent machine-learning approaches; so-called *empirical functionals* fit free parameters to experimental or compute-intensive high-accuracy reference data. Compared to contemporary ML-functionals, their parameter count is small; for example, the popular B3LYP functional by Becke (1993) has three open parameters ($a_0$, $a_x$, and $a_c$) that are least-squares fitted to atomization energies, ionization potentials, and proton affinities. In recent years deep learning has emerged as a promising direction toward improved DFAs. Beyond architectural advances, composite loss functions exemplify how ML practices have improved the field. Traditional empirical functionals have historically been overfitted to energies at the expense of accurate electron densities (Medvedev et al., 2017), a problem less prevalent in constraint-driven approaches (Kaplan et al., 2023). By jointly constraining energies and densities (Nagai et al., 2020), composite loss functions mitigate this failure mode.

**The parameterizations of most recent ML-DFAs** share a common pattern:

$$E_{\mathrm{xc}}[\rho] = \int \varepsilon_{\mathrm{xc}}^{\mathrm{base}}\big(\mathbf{g}_{\mathrm{local}}[\rho](\mathbf{r})\big)\, F_\theta\big(\mathbf{g}[\rho](\mathbf{r})\big)\rho(r)\, \mathrm{d}^3 r \,, \quad (6)$$

where $\varepsilon_{\mathrm{xc}}^{\mathrm{base}}$ is a traditional, computationally cheap approxi-

mation of the XC-energy density (e.g. energy density of the uniform electron gas), $\mathbf{g}_{\mathrm{local}}[\rho](\mathbf{r})$ is a vector of (semi)-local density features, e.g. $\mathbf{g}_{\mathrm{local}}[\rho](\mathbf{r}) = (\rho(r), |\nabla\rho(r)|, ...)$ (E.1), and $F_\theta$ is a learnable *enhancement factor* with parameters $\theta$. Typically, $F_\theta$ is modeled with a multi-layer perceptron (MLP) or compositions thereof. Depending on the architecture the density feature vector $\mathbf{g}[\rho](\mathbf{r})$ contains only local features (Dick & Fernandez-Serra, 2021; Nagai et al., 2022) or both local and non-local density features $\mathbf{g}[\rho](\mathbf{r}) = \mathbf{g}_{\mathrm{local}}[\rho](\mathbf{r}), \mathbf{g}_{\mathrm{non\text{-}local}}[\rho](\mathbf{r})$. The second category can be further subdivided into static predefined non-local density features (Nagai et al., 2020) and more expressive learnable non-local features (Gao et al., 2024; Luise et al., 2025). However, satisfactory basis-set generalization has not been demonstrated for these architectures, and it is reasonable to assume that evaluating them on a basis set family different from the one used during training would fail. Medvidović et al. (2025) proposed to learn the kinetic energy density $\tau : \mathbb{R}^3 \to \mathbb{R}$ to distill semi-local ($\mathcal{O}(N^3)$-scaling) models into cheaper local models of the same complexity class.

**Training XC-functionals** has proven to be challenging. Nagai et al. (2020) circumvented an auto-differentiable implementation altogether by falling back to stochastic parameter updates using Metropolis-Hastings type weight perturbation sampling. An alternative route of implementing a fully auto-differentiable SCF solver was first proposed by Tamayo-Mendoza et al. (2018). Li et al. (2021) showed that this end-to-end training approach has a regularizing effect resulting in improved generalization for one-dimensional hydrogen systems. Dick & Fernandez-Serra (2021) and later Gao et al. (2024) applied this training approach to cylindrical and arbitrary molecules, respectively. Kanungo et al. (2025) augment energy supervision with a constraint on the **ground-state** XC potential obtained via inverse DFT, penalizing the squared density-weighted potential error $\big(\int \rho(\mathbf{r})\, \Delta v_{\mathrm{xc}}(\mathbf{r})\, d\mathbf{r}\big)^2$.

## 4. Derivative Informed XC-Loss

We propose *Derivative Informed XC-Loss* (DI-Loss), which consists of two novel loss terms for learnable XC-functionals. It supervises the first and second derivatives of the energy with respect to the electron density on the Grassmann manifold of admissible, physically valid (idempotent) density matrices. The total loss function is given by a linear combination of energy $\mathcal{L}_E$ and density $\mathcal{L}_\rho$ terms, as well as our new gradient $\mathcal{L}_\nabla$ and Hessian $\mathcal{L}_H$ contributions

$$\mathcal{L}_{\mathrm{DI}} = \alpha_E \mathcal{L}_E + \alpha_\rho \mathcal{L}_\rho + \alpha_\nabla \mathcal{L}_\nabla + \alpha_H \mathcal{L}_H \quad (7)$$

where $\alpha_E, \alpha_\rho, \alpha_\nabla, \alpha_H$ are the loss weight constants for energy, density, gradient, and Hessian, respectively. We follow Li et al. (2021) and supervise the first three terms

$(\mathcal{L}_E, \mathcal{L}_\rho, \mathcal{L}_\nabla)$ along the densities of the SCF trajectory $\mathcal{L} = \sum_{j=0}^{N_{\text{cycles}}} \omega_j \mathcal{L}^{(j)}$ putting increasing weights $\omega_j$ on the later iterations. In the following, we denote the difference between a target quantity $X$ and its prediction $\hat{X}$ as $\Delta X := \hat{X} - X$.

For the **energy** loss $\mathcal{L}_E$ we use the mean squared error. For the **density** loss we use the per-electron $L^1$ norm of the real-space density error

$$\mathcal{L}_\rho^{(j)}[\rho, \hat{\rho}_j] = \frac{1}{N_{\text{elec}}} \int |\Delta\rho(\mathbf{r})| \, d\mathbf{r} \,, \qquad (8)$$

which performs best among the density loss types we compared (Appendix N).

The **gradient** $\boldsymbol{g} \in \mathbb{R}^{O \times V}$ with regard to the tangent coordinates (Appendix B) $\boldsymbol{\theta}_{ia} \in \mathbb{R}^{O \times V}$ of the Grassmann manifold

$$\boldsymbol{g}_{ia}^{(\text{xc})}[\rho] := \left. \frac{\partial E_{\text{xc}}[\rho(\mathbf{r}; \boldsymbol{\theta})]}{\partial \boldsymbol{\theta}_{ia}} \right|_{\boldsymbol{\theta}=0} \qquad (9)$$

can be efficiently computed in closed-form (Appendix B)

$$\boldsymbol{g}_{ia}^{(\text{xc})} = -2(\boldsymbol{C}^T \boldsymbol{V}_{\text{xc}} \boldsymbol{C})_{ia} \,. \qquad (10)$$

The gradient of the total energy is given by

$$\boldsymbol{g}_{ia}^{(\text{tot})}[\rho] := \frac{\partial E_\rho[\rho(\mathbf{r}; \boldsymbol{\theta})]}{\partial \boldsymbol{\theta}_{ia}} + \boldsymbol{g}_{ia}^{(\text{xc})} \,, \qquad (11)$$

such that the mean-field term $E_\rho$ cancels out in the loss computation and we obtain

$$\mathcal{L}_\nabla^{(j)} = \frac{1}{N_{\text{elec}}} \left\| \left( \left( \boldsymbol{C}^{(j)} \right)^T \boldsymbol{\Delta} V_{\text{xc}}^{(j)} \boldsymbol{C}^{(j)} \right)_{ia} \right\|_F \,. \qquad (12)$$

In contrast to this construction, Kanungo et al. (2025) match the potential through a single density-weighted scalar per sample, $\left( \int \rho(\mathbf{r}) \Delta v_{\text{xc}}(\mathbf{r}) \, d\mathbf{r} \right)^2$. This fixes only the $\rho$-weighted mean of the potential error and leaves every variation with $\int \rho \, \delta v = 0$ unconstrained. Supervising the full $V_{\text{xc}} \in \mathbb{R}^{B \times B}$ matrix goes to the opposite extreme, since its occupied-occupied and virtual-virtual blocks describe rotations that leave the density invariant, forcing the model to spend capacity on potential structure with no physical effect. Our gradient instead constrains the $O \times V$ occupied-virtual block, the directions that move the density and drive the SCF update, giving $O \cdot V$ independent constraints per sample and SCF cycle. It supervises these directions along the trajectory rather than at the ground state alone. Kanungo et al. (2021) show that small density errors correspond to large potential errors, so matching them away from equilibrium is a stronger requirement than matching the ground-state density. Early ablations showed that supervising the full $V_{\text{xc}}$ matrix degrades distillation accuracy relative to the Grassmannian

gradient, consistent with the model wasting capacity on the unphysical occupied-occupied and virtual-virtual blocks.

The **Hessian** is only supervised at the equilibrium density to focus model expressivity on the physically relevant curvature of the energy functional. The Hessian at the ground state governs SCF stability and linear response, while second-order information far from the minimum is weakly constrained and unnecessary for many practical applications. Even though the dimensionality of the Grassmann manifold is much smaller than that of the full $B \times B$ matrix space, the corresponding Grassmannian Hessian

$$\boldsymbol{H}_{iajb} = \left. \frac{\partial^2 E_{\text{xc}}[\rho(\mathbf{r}; \boldsymbol{\theta})]}{\partial \boldsymbol{\theta}_{ia} \partial \boldsymbol{\theta}_{jb}} \right|_{\boldsymbol{\theta}=0} \qquad (13)$$

remains prohibitively large to materialize. Instead, we supervise randomly sampled linear responses

$$\delta\boldsymbol{g}_{\mu i}^{(\text{xc})} = \left. \frac{\partial \boldsymbol{g}_{\mu i}^{(\text{xc})}[\rho(r; \boldsymbol{\theta})]}{\partial \boldsymbol{\theta}_{\nu j}} \right|_{\boldsymbol{\theta}=0} \delta\boldsymbol{\theta}_{\nu j} \qquad (14)$$

which are efficiently computable using Hessian-vector products (Appendix D). Perturbation directions are sampled with importance weights biased toward small occupied-virtual gaps

$$\delta\theta_{ia} \propto z_{ia}/(\epsilon_a - \epsilon_i), \quad z_{ia} \sim \mathcal{N}(0, 1) \,. \qquad (15)$$

This weighting focuses the Hessian supervision on the low-gap transitions that dominate linear response and TDDFT excitation energies. We define the Hessian loss as the Monte Carlo expectation

$$\mathcal{L}_H = \mathbb{E}_{\delta\boldsymbol{\theta}} \left[ \frac{\|\Delta\delta\boldsymbol{g}^{(\text{xc})}\|_F^2}{\|\delta\boldsymbol{g}^{(\text{xc})}\|_F^2} \right] \,, \qquad (16)$$

where we normalize by the target response magnitude to aid training stability, approximated with $T = 8$ samples per training step.

The **gap weighting** (15) carries a precise physical meaning beyond down-weighting high-gap transitions. We collect the gaps into a diagonal matrix $\boldsymbol{D}$ with entries $d_{ia} = \varepsilon_a - \varepsilon_i$, so the sampled direction is $\delta\boldsymbol{\theta} = \boldsymbol{D}^{-1}\boldsymbol{z}$. The same $\boldsymbol{D}^{-1}$ plays three roles. First, as the leading-order inverse orbital Hessian it preconditions the SCF iteration, since a first-order update from the occupied-virtual Fock residual scales as $\theta_{ia} \propto -F_{ia}/(\varepsilon_a - \varepsilon_i)$ (Helgaker et al., 2000). Second, it is proportional to the static Kohn-Sham response $\chi_s$, diagonal in the particle-hole basis with entries $\propto d_{ia}^{-1}$ (Appendix G), which sets the metric of the OEP equation $\chi_s \hat{V}_{xc}^{\text{OEP}} = \Lambda$. Third, it inverts the independent-particle block of the Casida matrix $\boldsymbol{A} = \boldsymbol{D} + \boldsymbol{K}$ (Appendix F), where the supervised Hessian is the exchange-correlation ($f_{xc}$) part of the coupling block $\boldsymbol{K}$. The sampling therefore draws $\delta\boldsymbol{\theta} \propto \chi_s \boldsymbol{z}$, the first-order response to a random potential perturbation,

equivalently the SCF rotation induced by a random Fock residual. The perturbations concentrate where the response function, the SCF iteration, and the OEP residual are jointly most sensitive. This sampling yields an OEP-inspired curvature match at the cost of one batched Hessian-vector product, without forming or inverting the response operator $\chi_s$ that the OEP equation requires.

The four terms carry complementary information about the ground-state energy basin. Energy supervision fixes the scalar value at the minimum and leaves the density that attains it unconstrained. Density supervision locates the minimum but not the path toward it. Gradient supervision adds the local slope, the direction of steepest descent in density space. Hessian supervision adds the curvature, which governs convergence and links to linear response. The training signal moves from the value at a single point to the shape of the basin around it.

# 5. Experiments

**Datasets:** We train and test on QM9 molecules (Ramakrishnan et al., 2014) and use QM40 (Madushanka et al., 2024) for far out-of-distribution evaluation. We split QM9 by heavy-atom count, training and validating on the smaller molecules and testing on the larger ones to isolate size extrapolation. From QM9 we exclude fluorine, since it is rare in the dataset and the heavy-atom split leaves close to none in the training set. From QM40 we exclude fluorine, sulfur, and chlorine, since these elements are scarce or absent in QM9 and would otherwise add element extrapolation to the size-extrapolation test. We subsample QM40 to 50 molecules per heavy-atom bin up to 40 heavy atoms.

**Evaluation metrics:** Total energy alone does not capture functional quality, especially when directly optimized. We therefore evaluate a diverse set of properties: total energy $E_{\text{tot}}$, energy components ($E_\rho$, $E_C$, $E_{\text{xc}}$), HOMO-LUMO gap $\varepsilon_{\text{HL}}$, TDDFT excited-state energies, density errors ($L_1$, $L_2$, dipole $\mu_\rho$), and SCF convergence behavior. See Appendix J for details.

**Adaptive Training Stabilization:** Training ML-XC functionals end-to-end through the SCF solver is challenging because parameter updates that destabilize the XC-potential prevent proper SCF convergence, producing unreliable gradients that further degrade subsequent updates. Viewed as a fixed-point iteration $P^{(t+1)} = f_\theta(P^{(t)})$, where the map $f_\theta$ is implicitly defined by the learned XC functional, SCF can be connected to a setting studied in the literature as *Deep Equilibrium Models* (DEQs) (Bai et al., 2019). Bai et al. (2021) pointed out that such architectures are indeed brittle and addressed their instability by regularizing the fixed-point map Jacobian, but their approach does not transfer directly to the SCF setting. Our Hessian supervision can

be viewed as an alternative to this, as it encodes the ground state as a stable attractor of the SCF-map.

However, to reliably compare all loss settings and prevent individual bad updates from deraining the training, we employ a Metropolis-inspired accept-reject mechanism based on an adaptive variance estimate of the relative mean epoch loss change (Appendix H). Updates exceeding a given tolerance are rejected and the optimizer momentum is rescaled after consecutive rejections. This stabilization scheme enables a simplified single-stage, gradient-based training procedure, unlike the multi-stage approaches of Dick & Fernandez-Serra (2021), Kirkpatrick et al. (2021), and Luise et al. (2025), or the gradient-free optimization of Nagai et al. (2020). Notably, this procedure converges reliably from standard MINAO density initializations, without requiring the pre-converged densities used in prior work (Dick & Fernandez-Serra, 2021; Gao et al., 2024).

**Parameter Optimization and Initialization:** Since DEQs are known to be sensitive w.r.t. their initialization (Agarwala & Schoenholz, 2022), we calibrate the initial weight distribution to be variance preserving across layers[2]. More generally, we observe significant improvements using the Muon optimizer (Jordan et al., 2024) compared to Adam (Kingma & Ba, 2017) across all loss configurations (Appendix I) and use it throughout. All runs are performed with different random seeds sampled from hardware entropy.

**Models:** We evaluate DI-Loss on multiple ML-XC architectures. NNmGGA (Nagai et al., 2020) is a simple constraint-free neural network operating on semi-local density features. The semi-local model by Dick & Fernandez-Serra (2021) explicitly enforces physical constraints through architecture design similar to traditional constraint-based DFAs (Sun et al., 2015). EG-XC (Gao et al., 2024) augments learnable mGGA functionals to non-local models via equivariant message passing on nuclei-centered representations. In our distillation experiments, we use NNmGGA as the local component of EG-XC. Skala (Luise et al., 2025) is broadly similar to EG-XC but without message passing on the nuclei-centered representations; to isolate the influence of their grid-based MLP architecture, we focus on *Skala mGGA*, their model with the non-local augmentation disabled[3]. To compare between models, we match the semi-local model sizes within each distillation setting (Appendix M).

## 5.1. Learning mGGAs

To investigate the influence of DI-Loss terms we start with a toy example. We learn an ML mGGA representation of a

---

[2]We do not do this for Skala since its reference implementation explicitly pairs uniform Xavier initialization with SiLu-activations

[3]This is equivalent to setting the non-local flag to off in their reference implementation

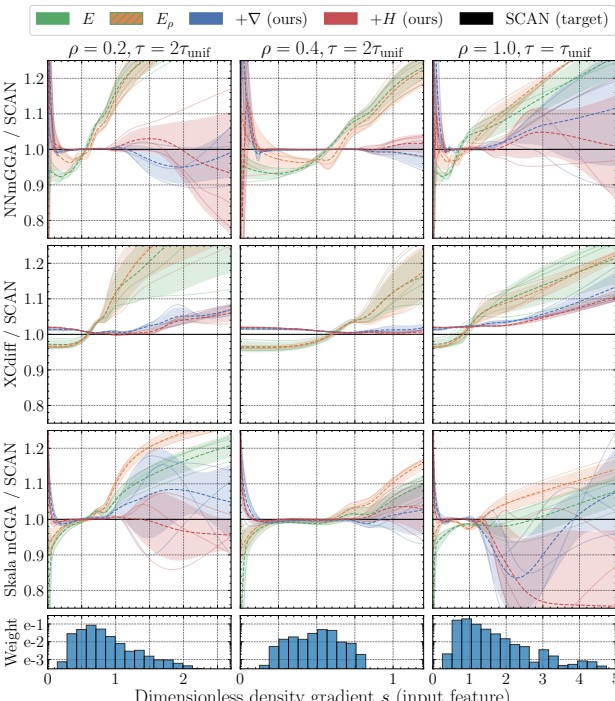

*Figure 1.* **Diagnostic toy example: energy-density of mGGA functionals distilled from SCAN.** Each panel plots the ratio of learned to target energy density (y-axis) against the dimensionless density gradient $s$ (x-axis), a standard semi-local input feature (Appendix E.1). Rows are the three architectures, columns three density regimes set by $(n, \tau)$, and a perfect functional traces the line at one. The bottom row shows the importance-weighted ($\omega_{\text{quadrature}} \times \rho \times |\varepsilon_{\text{target}}|$) distribution of $s$ on an ethanol quadrature grid, marking where errors carry energetic weight. Energy supervision deviates from the target even in the data-rich region. The gradient and Hessian terms ($+\nabla$, $+H$) remove these deviations and extrapolate into low weight grid feature regimes. Supervising only energy-type quantities, this setting isolates how the derivative terms reshape the energy density without any density loss.

traditional mGGA, where the learned and target functionals share the energy-density form $\varepsilon_{\text{xc}}(\rho, \zeta, s, \tau)$ and we can inspect the learned functional directly. Training against a reference in the same XC category keeps model expressivity from confounding the loss-term influence (Sun et al., 2015). We scale all semi-local models to the same parameter count (Appendix M), which gives approximately equal cost per SCF step. We train on QM9 molecules with up to 5 heavy atoms, which leaves only 174 molecules. To isolate the derivative terms we supervise only energy-type quantities, the total energy, the mean-field energy $E_\rho$ in place of the tuned $L_1$ density loss, and the energy gradient and Hessian. This handicaps the baseline on purpose, since the point is to read off what the gradient and Hessian add when no real-space density signal is present.

Crucially, the mGGA-to-mGGA setting provides a diagnostic in which the learned functionals can be analyzed at the level of energy densities $\varepsilon_{\text{xc}}(\rho, \zeta, s, \tau)$. Figure 1 demon-

strates that DI-Loss systematically corrects energy misallocation (in the absence of density supervision) across density regimes, providing insight into where and how the learned functional improves, rather than treating it as a black box. Table 6 shows the same effect on the energy components, where the gradient term reduces the Coulomb and exchange-correlation errors that energy-only supervision leaves two orders of magnitude off.

## 5.2. Distillation of XC-Functionals

We further evaluate DI-Loss in a distillation setting, compressing B3LYP ($\mathcal{O}(N^4)$) into semi-local functionals ($\mathcal{O}(N^3)$) and testing out-of-distribution on QM40 (Figure 2). Since hybrid functionals are fundamentally non-local and cannot fulfill the same set of constraints as semi-local methods, we swap out the constraint-based XCdiff architecture (Dick & Fernandez-Serra, 2021) for the non-local EG-XC functional using NNmGGA as a local model (Gao et al., 2024; Nagai et al., 2020). The computational scaling of DFT is more accurately described in terms of the basis size $B$ rather than atom count, with the commonly cited $\mathcal{O}(N^X)$ behavior reflecting scaling in $B$, which itself grows linearly with system size for fixed basis set families. To reduce the cost of the $\mathcal{O}(B^4)$-scaling hybrid reference, we use the smaller def2-SVP basis set for these experiments. We use moderately larger local networks than in the SCAN experiments (Appendix M), since the non-local character of exact exchange places greater demands on model capacity than semi-local distillation.

DI-Loss lowers the total energy error of every architecture and leaves the density metrics near the $E + \rho$ baseline. The effect is largest on EG-XC, where the full loss cuts the total energy error by $80\%$ and is the only architecture whose density also improves, lowering $L_2[\rho]$ by $8\%$. We attribute this to its non-local representation having the capacity to absorb the derivative signal that the semi-local models cannot, though we have trained EG-XC only once per setting and cannot rule out seed variance. The distilled functionals require $10$–$15\%$ more SCF iterations than the B3LYP reference, and the Hessian term recovers part of this overhead, consistent with our DEQ-regularization hypothesis that curvature supervision regularizes the energy landscape near the ground state.

## 5.3. Accelerated-SCF using ML-XC-Functionals

.5 Figure 2 shows that the self-consistent densities of the distilled functionals remain close to the B3LYP optimum in an energetic sense. When evaluated with the B3LYP functional, these densities yield errors below 1 mEh on the out-of-distribution QM40 test set: 0.85–0.9 mEh for the $E + \rho$ baselines and 0.67 mEh with DI-Loss. The gain from derivative supervision is modest for this metric, but the absolute er-

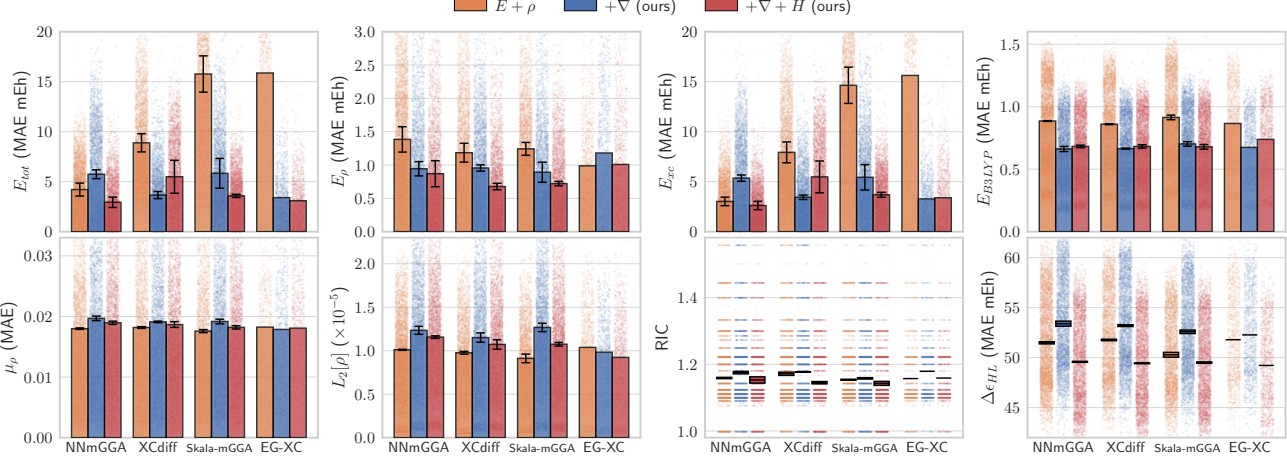

*Figure 2.* **Distillation of $\mathcal{O}(N^4)$-scaling B3LYP into $\mathcal{O}(N^3)$-scaling ML-XC functionals.** We train on QM9 molecules with up to 7 heavy atoms and evaluate out-of-distribution on a random QM40 subsplit, fixed across all models, at the B3LYP/def2-SVP level of theory. Bars give the mean MAE over random seeds, error bars the standard error of that mean, and the jittered dots resolve the per-molecule errors that the means average over. DI-Loss sharpens the energetic quantities ($E_{\text{tot}}$, $E_{\text{xc}}$) and leaves the density metrics ($\mu_\rho$, $L_2[\rho]$) near the $E + \rho$ baseline across the three semi-local architectures. On the more expressive EG-XC architecture the added terms improve the densities, lowering $L_2[\rho]$, whereas they slightly worsen it on the three semi-local architectures. We train EG-XC once per setting, so we treat this as suggestive rather than conclusive. Full per-run numbers are in Table 7.

ror is small. This observation has motivated us to explore the use of these densities as initial guesses for the B3LYP functional as a secondary application for distilled functionals beyond direct evaluation. We run the lower-scaling distilled functional and use the resulting density as the initial guess for a subsequent B3LYP calculation. This follows a standard electronic-structure strategy, where cheaper methods initialize more expensive calculations (Stein & Hutter, 2022), but applies it to distilled neural network XC functionals.

Figure 3 shows that this results in a significant reduction in the required number of solver iterations relative to standard MINAO initialization for B3LYP. The traditional semi-local initialization baseline reduces the relative iteration count to about 65%, while distilled ML-XC densities reduce it further to about 50% across all architectures. This acceleration effect is largely unaffected by DI-Loss. Distillates trained only with $E + \rho$ already provide low-RIC initializations, and derivative supervision changes RIC only marginally relative to the gain from distillation itself.

RIC reduction alone does not imply walltime reduction, since the distilled pre-run adds additional SCF cycles. We therefore measure the end-to-end runtime of ML-XC-initialized B3LYP in Appendix K. On a QM40 molecule with 35 heavy atoms, six NNmGGA cycles followed by the main B3LYP calculation results in a $1.35\times$ walltime speedup. This finite-size gain should increase for larger systems and larger basis sets, since the distilled pre-run avoids the $\mathcal{O}(B^4)$ exact-exchange step of B3LYP. For a fixed number of ML-XC pre-run cycles, the relative cost

of the initialization therefore decreases with basis size. In this regime, the walltime reduction approaches the RIC reduction, and the speedup factor approaches $1/\text{RIC}$. Appendix K provides single-cycle benchmarks supporting this

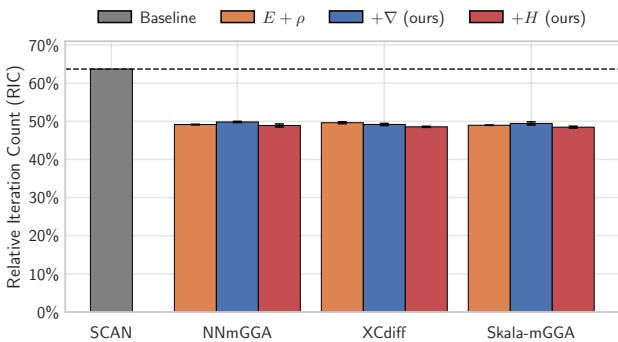

*Figure 3.* **Relative iteration count (RIC) for B3LYP** calculations initialized from SCAN or distilled ML-XC densities on QM9, trained on molecules with up to 7 heavy atoms and evaluated on 1000 randomly selected molecules with exactly 9 heavy atoms, fixed across all runs. RIC is the number of B3LYP iterations required from a given initial density, normalized by standard MINAO initialization. SCAN reduces the iteration count to about 65% of MINAO, while distilled ML-XC densities reduce it further to about 50% across all architectures. Hence functional distillation provides a secondary use as initializers for subsequent calculations with more expensive traditional functionals. Here derivative supervision has only a minor effect on RIC. DI-Loss supervision gives the lowest mean RIC, but the gain over $E + \rho$ is small relative to the gain from distillation itself.

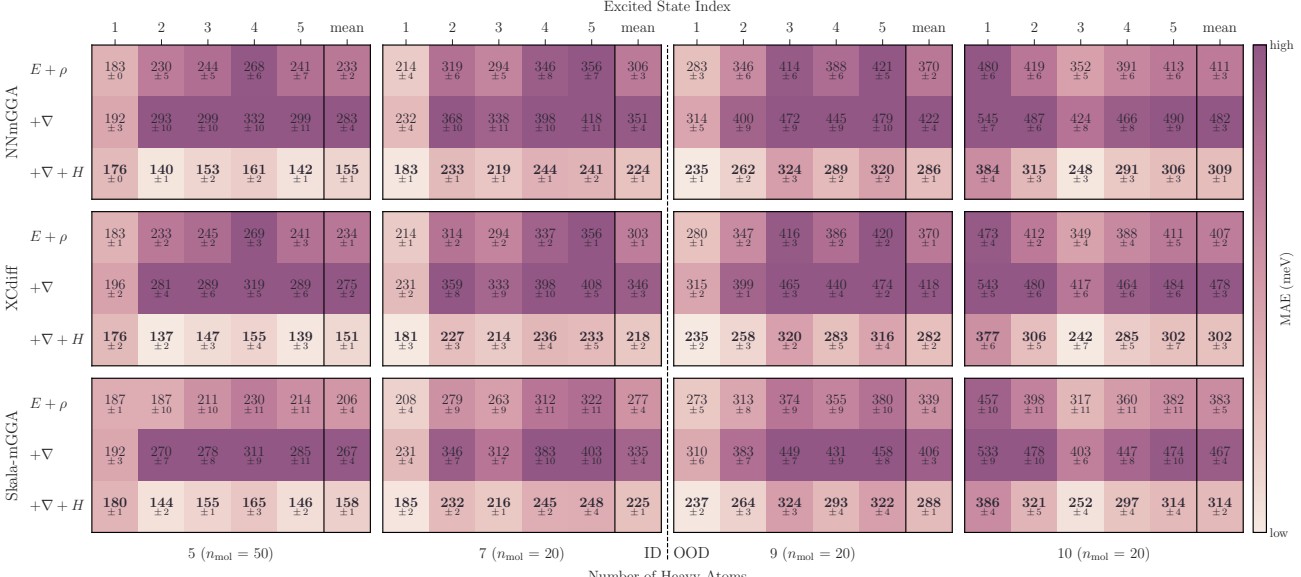

*Figure 4.* **Hessian supervision improves TDDFT excited-state accuracy across functionals and molecule sizes.** Each cell reports the mean absolute error (MAE, meV) of the lowest five vertical excitation energies against B3LYP, averaged over $n_{mol}$ molecules at each heavy-atom count. Rows group the three neural functionals (NNmGGA, XCdiff, Skala-mGGA). Within each functional, the three sub-rows add training signal cumulatively, from energy and density ($E + \rho$), to first derivatives ($+\nabla$), to the full DI-Loss with the Hessian ($+\nabla + H$). Columns give the excited states (1–5) and their mean MAE. The dashed line separates in-distribution QM9 molecules from larger out-of-distribution molecules from QM9 and QM40. Color encodes the MAE (light low, dark high, normalized within each heavy atom column), and ± values are the standard error across different random seeds. Hessian supervision lowers MAE across every functional and size. In-distribution, the first excitation already carries the lowest error (176 meV, XCdiff, 5 heavy atoms), so the gains concentrate on the higher states, where the DI-Loss cuts the energy MAE of the fifth excitation from 241 to 139 meV (XCdiff, 5 heavy atoms). Out-of-distribution, this lowest state also improves (473 to 377 meV, XCdiff, 10 heavy atoms). XCdiff gives the lowest mean MAE at every size. Relative to $E + \rho$, the DI-Loss reduces the XCdiff mean by 35% (5 heavy), 28% (7 heavy), 24% (9 heavy), and 26% (10 heavy atoms).

scaling trend.

Prior machine-learned density (Song & Feng, 2024) and Hamiltonian (Yu et al., 2023) predictors have also been used for SCF initialization. These methods are cheaper than our approach, since they predict an initial density or Hamiltonian in a single forward pass rather than running a distilled SCF calculation. However, matrix predictors trained only on converged ground states fail at size extrapolation (Liu et al., 2025). Our approach makes a different tradeoff. It adds the cost of several SCF cycles with the distilled functional, but gives a larger iteration reduction in this setting. On QM40 at the same B3LYP/def2-SVP level, solver-aligned matrix prediction reduces the B3LYP iteration count by about 20–30% (Eberhard et al., 2026), while distilled functional initialization reduces it by about 50%. So these two approaches target different cost regimes. Matrix predictors provide cheaper initial guesses, while distilled functionals provide stronger initialization when the additional pre-run cost is amortized by the computational scaling of the hybrid calculation.

## 5.4. TDDFT

The orbital Hessian that governs the linear-response equations is the second derivative that the DI-Loss supervises, so we expect the Hessian term to improve the excitation energies. We compute the lowest five vertical excitations for each trained functional and report the MAE against the B3LYP target in Figure 4, across two in-distribution sizes (5, 7 heavy atoms) and two out-of-distribution sizes (9, 10 heavy atoms).

Adding only the gradient term raises the MAE above the $E + \rho$ baseline for every functional and size (13–18% for XCdiff). The Hessian term reverses this and lowers the MAE below the baseline at every size. In-distribution, the first excitation is already accurate, so the reduction concentrates on the higher states. Out-of-distribution, the first state benefits as well. XCdiff reaches the lowest mean error throughout, where the DI-Loss cuts the mean by 35% at 5 heavy atoms and 26% at 10.

# 6. Discussion

In this work, we introduced *Derivative Informed XC-Loss* (DI-Loss), a composite loss function that regularizes ML-XC functionals by supervising derivatives of the energy on the Grassmannian of admissible density matrices. We evaluate this objective in a hybrid-distillation setting, where semi-local ML-XC functionals are trained to reproduce B3LYP/def2-SVP targets while retaining the lower scaling of non-hybrid functionals. In this setting, DI-Loss consistently improves the main energy metrics across all four evaluated architectures. For the selected loss weights, the total-energy MAE decreases by $66\%$ when averaged uniformly across architectures. These improvements are strongest for Skala-mGGA and EG-XC, where the total-energy MAE decreases from 15.8 to 3.6 mEh and from 15.9 to 3.1 mEh, respectively. NNmGGA and XCdiff also improve, with total-energy MAEs decreasing from 4.2 to 2.9 mEh and from 8.9 to 5.5 mEh. The effect on density-related quantities is more metric-dependent. The mean-field energy metric $E_\rho$, which has been proposed as a density-sensitive benchmark by Gould (2023), improves on average from 1.2 to 0.8 mEh. In contrast, the direct density metrics $\mu_\rho$ and $\mathcal{L}_2[\rho]$ do not improve uniformly.

We show that the response information introduced by Hessian supervision improves downstream excited-state calculations. In TDDFT calculations against B3LYP reference excitations, adding Hessian supervision lowers the mean excitation-energy MAE across functionals and molecule sizes. For the strongest architecture, XCdiff, the full DI-Loss reduces the mean MAE relative to $E + \rho$ training by $35\%$, $28\%$, $24\%$, and $26\%$ for molecules with 5, 7, 9, and 10 heavy atoms, respectively. In-distribution, where the first excitation is already comparatively accurate, the gains are concentrated on higher excited states. For example, the MAE of the fifth excitation decreases from 241 to 139 meV for XCdiff on molecules with 5 heavy atoms. Out-of-distribution, the first excitation also improves, decreasing from 473 to 377 meV for XCdiff on molecules with 10 heavy atoms.

Overall, these results indicate that Grassmannian derivative supervision improves more than the fitted ground-state energy. It regularizes the learned functional in directions that affect the self-consistent density, orbital gaps, and the response properties entering downstream TDDFT calculations.

**Limitations and Future Work.** The additional loss terms we propose are only directly applicable to the distillation setting, where reference gradients and Hessians are readily available from a reference functional. Extending DI-Loss to higher-fidelity targets such as CCSD(T) would require inverse DFT techniques to obtain the corresponding XC potential and its derivatives, which remains computationally de-

manding (Kanungo et al., 2025). Nevertheless, we envision a multi-stage training strategy analogous to recent advances in machine-learned interatomic potentials (MLIPs), where combining lower-fidelity pretraining with higher-fidelity fine-tuning improves performance at reduced computational cost (Kulichenko et al., 2024). Concretely, DI-Loss could enable efficient pretraining on range-separated or double-hybrid functionals, which currently compete with $\mathcal{O}(N^3)$-scaling ML-XC functionals (Luise et al., 2025), followed by fine-tuning on CCSD(T) energies or inverse-DFT-derived XC potentials. For double-hybrid distillation in particular, correct orbital energies are essential, and density-only supervision leaves virtual orbital energies largely unconstrained, making our Hessian supervision especially relevant.

All experiments in this work are restricted to closed-shell organic molecules, a relatively narrow chemical domain. While our molecular runtime benchmarks already show favorable scaling compared to B3LYP (Appendix K), hybrid-functional acceleration is likely even more consequential in solid-state settings where exact exchange is substantially more expensive. Whether the benefits of DI-Loss transfer to periodic solid-state systems remains an open question. Additionally, while Hessian supervision improves the quality of the learned functional, it introduces a moderate training cost overhead (Appendix K.2). However, this overhead is confined entirely to training, and the resulting semi-local functional incurs no additional inference cost.

A further open question concerns the expressivity of current ML-XC architectures. Our evaluations suggest that even recent $\mathcal{O}(N^3)$-scaling parameterizations may lack the capacity to fully capture exact exchange effects. Whether this expressivity gap can be closed without sacrificing computational efficiency remains unclear. However, it is also unknown whether the true XC functional exhibits such hybrid-like features that resist approximation by $\mathcal{O}(N^3)$ architectures, and this limitation may prove less relevant when training on wavefunction-based targets such as CCSD(T). Additionally, we note that our architectural comparisons were performed on down-scaled versions of the original architectures and larger-scale ablations would strengthen expressivity claims.

More broadly, beyond the performance gains reported here, our robust single-stage training procedure enables cross-architecture comparisons under consistent protocols, addressing a key evaluation gap in prior ML-XC work. Finally, KS-DFT serves as a reference theory for generating large-scale datasets used to train machine-learned force fields (Levine et al., 2025), and improvements in XC-functional accuracy have the potential to indirectly enhance the fidelity of these downstream models, amplifying the impact of improved density functional approximations.

## Acknowledgements

We thank Jacobus Dijkman for insightful discussions regarding Hessian supervision in classical density functional theory.

## Impact Statement

This work develops training strategies for ML-XC-functionals aiming to improve the accuracy and stability of electronic structure simulations. By enabling more reliable and efficient quantum chemical calculations, the proposed methods may contribute to advances in computational chemistry workflows used in materials science, molecular modeling, and chemical design. Potential downstream applications include accelerated screening of functional materials, improved modeling of molecular properties, and more efficient electronic-structure simulation in academic and industrial research. We do not foresee societal risks beyond those generally associated with advances in chemical simulation and materials modeling.

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

# A. KS-DFT in Finite Basis Sets

In practice, the minimization of the Kohn-Sham functional (Eq. 1) is done in a finite basis set using an iterative solver. In this section we closely follow Lehtola et al. (2020) and the notation used therein to provide a brief overview of the solution process, for more detailed explanations we recommend reading their paper. Note that we further simplify by removing any reference to spin, making our summary here more accessible to a wider audience and implicitly adhering to the spin-restricted case and an even number of electrons. For notational convenience, we employ Einstein's summation convention, where repeated indices imply summation over their range.

In this work, we use an atom-centered *Gaussian-type* basis set $\mathbf{X} := \{\chi_\mu | \chi_\mu : \mathbb{R}^3 \to \mathbb{R}\}_{\mu=1}^B$. The overlap between basis functions is defined as the spatial integral of their product. The corresponding *overlap matrix* $S$ stores all of these pairwise integrals

$$S_{\mu\nu} := \int_{\mathbb{R}^3} \chi_\mu(r)\chi_\nu(r)\, \mathrm{d}r \,. \tag{17}$$

Notably, the basis set is not orthogonal and hence $S$ is not diagonal. The motivation to use Gaussians to construct $\mathbf{X}$ is that most integrals like these can be computed very efficiently using closed-form solutions.

In KS-DFT, the electron density $\rho : \mathbb{R}^3 \to [0, \infty)$ is represented by a set of *molecular orbitals* (MOs) $\{\varphi_i | \varphi_i : \mathbb{R}^3 \to \mathbb{R}\}_{i=1}^B$ which are the eigenfunctions of (eq. 2). These orbitals can be represented by coefficient vectors $c^{(i)} \in \mathbb{R}^B$ in a given basis set $\varphi_i(r) = c_\mu^{(i)}\chi_\mu(r)$ The full set of orbitals can conveniently be represented in a matrix $C = (c^{(1)}, c^{(2)}, ..., c^{(B)}) \in \mathbb{R}^{B \times B}$, where the leading index is used for the basis $\mathbf{X}$ and the second one indexes the MOs. We follow the convention to use Greek letters to index the basis functions and Roman letters for the MOs. Unlike the basis set, the MOs are orthonormal.

$$\delta_{ij} = \int_{\mathbb{R}^3} \varphi_i(r)\varphi_j(r)\, \mathrm{d}r = C_{i\mu}C_{j\nu} \int_{\mathbb{R}^3} \chi_\mu(r)\chi_\nu(r)\, \mathrm{d}r =: C_{i\mu}C_{j\nu}S_{\mu\nu}. \tag{18}$$

While the expansion of the MOs is linear, the resulting expansion of the electron density is quadratic in the basis functions

$$\rho(r) = 2 \sum_i^{N_e/2} |\varphi_i(r)|^2 = \chi_\mu(r)\chi_\nu(r) \underbrace{2C_{\mu i}C_{\nu i}}_{=:P_{\mu\nu}} \,, \tag{19}$$

where $P \in \mathbb{R}^{B \times B}$ is the so-called density matrix. An instructive example of the expressivity of this discretization is that the integral over the density simplifies to a sum over $B^2$ matrix elements $\int_{\mathbb{R}^3} \rho(r)\, \mathrm{d}r = P_{\mu\nu}S_{\mu\nu}$. The total electronic energy in a finite basis set is given by (Lehtola et al., 2020)

$$E_{\mathrm{tot}} = P_{\mu\nu}H_{\mu\nu}^{(\mathrm{core})} + \frac{1}{2}P_{\mu\nu}(\mu\nu|\lambda\sigma)P_{\lambda\sigma} + E_{\mathrm{xc}}[\rho(P; \mathbf{X})] \,, \tag{20}$$

with the *core Hamiltonian* ($\in \mathbb{R}^{B \times B}$)

$$H_{\mu\nu}^{(\mathrm{core})} = \int_{\mathbb{R}^3} \chi_\mu(r) \left(-\frac{1}{2}\nabla^2 + \sum_n \frac{Z_n}{|r_n - r_1|}\right) \chi_\mu(r)\, \mathrm{d}r_1 \,, \tag{21}$$

and the *electron repulsion integral* (ERI) ($\in \mathbb{R}^{B \times B \times B \times B}$) tensor

$$(\mu\nu|\lambda\sigma) := \iint_{\mathbb{R}^3} \chi_\mu(r_1)\chi_\nu(r_1) \frac{1}{|r_1 - r_2|} \chi_\lambda(r_2)\chi_\sigma(r_2)\, \mathrm{d}r_1\mathrm{d}r_2 \,. \tag{22}$$

Both the core Hamiltonian and the ERI tensor are constant w.r.t. the electron density, and only depend on the basis set[4]. The derivative of the energy $E_{\mathrm{tot}}$ w.r.t the density matrix $P$ is commonly referred to as the *Fock matrix*

$$F_{\mu\nu} := \frac{\partial E_{\mathrm{tot}}}{\partial P_{\mu\nu}} = H_{\mu\nu} + (\mu\nu|\lambda\sigma)P_{\lambda\sigma} + \frac{\partial E_{\mathrm{xc}}}{\partial P_{\mu\nu}}, \tag{23}$$

---

[4]which in turn depends on the nuclear point cloud

where the last term is the discretized *XC-potential* denoted as $V_{\mu\nu}^{(\mathrm{xc})}$. Its continuous counterpart $v_{\mathrm{xc}} : \mathbb{R}^3 \to \mathbb{R}$ appears in Equation 2, these relate to each other via

$$V_{\mu\nu}^{(\mathrm{xc})} = \int \chi_\mu(\mathbf{r}) v_{xc}(\mathbf{r}) \chi_\nu(\mathbf{r}) \, d\mathbf{r} \, . \tag{24}$$

One can show (see, for example, Lehtola et al. (2020)) that the discretized problem of minimizing Equation (1) with respect to the density leads to the Roothaan–Hall equations

$$F(P(C)) \, C = \mathrm{diag}(\varepsilon_1, \dots, \varepsilon_B) \, SC \, , \tag{25}$$

where the eigenvalues $\varepsilon$ correspond to the Kohn–Sham orbital energies and $C \in \mathbb{R}^{B \times B}$ contains the orbital coefficients. We explicitly indicate the dependence of the Fock matrix $F$ on the density matrix $P$ through the density dependence of the exchange-correlation potential $V^{(\mathrm{xc})}$, highlighting that (25) constitutes a nonlinear eigenvalue problem. In practice, this problem is solved iteratively by linearizing the equations through fixing $F$, solving the resulting generalized eigenvalue problem to obtain an updated density matrix $P$, and recomputing $F$ from it. Repeating this procedure until convergence yields a self-consistent solution. Consequently, finding the ground-state density can be interpreted as a fixed-point problem in the space of density matrices, or equivalently Fock matrices, satisfying

$$F^* = \mathrm{SCF}_{\mathrm{step}}(F^*) \, . \tag{26}$$

## B. Orbital Rotations and Direct Minimization

It can be shown that the density matrix $P$ is invariant under orthonormal linear transformations $\mathbf{O}_N(\mathbb{R}) = \{U \in \mathbb{R}^{N \times N} | U^T U = U U^T = \mathbf{1}\}$ of the occupied and virtual submatrices of the coefficient matrix

$$C = \begin{pmatrix} C_\mathrm{o} & C_\mathrm{v} \end{pmatrix} , \tag{27}$$

where $C_\mathrm{o} \in \mathbb{R}^{O \times B}$ and $C_\mathrm{v} \in \mathbb{R}^{V \times B}$ (Lehtola et al., 2020). Meaning that

$$\forall \, U_{\mathrm{oo}} \in \mathbf{O}_\mathrm{o}(\mathbb{R}), \, U_{\mathrm{vv}} \in \mathbf{O}_\mathrm{v}(\mathbb{R}) \text{ and } C' := C \begin{pmatrix} U_{\mathrm{oo}} & 0 \\ 0 & U_{\mathrm{vv}} \end{pmatrix} : \, P(C') = P(C) \, . \tag{28}$$

$$C(\theta) := C_0 \exp \begin{pmatrix} 0 & \theta_{\mathrm{ov}} \\ -\theta_{\mathrm{ov}}^T & 0 \end{pmatrix} \tag{29}$$

In the following, we use $i, j, \dots \in \{1, \dots, O\}$ and $a, b, \dots \in \{O+1, \dots, B\}$ to index occupied and virtual orbitals, respectively

$$\left. \frac{\partial E}{\partial \theta_{ia}} \right|_{\theta=0} = \frac{\partial E}{\partial P_{\eta\zeta}} \frac{\partial P_{\eta\zeta}}{\partial \theta_{ia}} = -F_{\eta\zeta} \left[ C_{\eta a} C_{\zeta i} + C_{\eta i} C_{\zeta a} \right] = -2(C^T F C)_{ia} =: -2\tilde{F}_{ia} \tag{30}$$

Analogously, we can derive that $\frac{\partial E_{\mathrm{xc}}}{\partial \theta_{ia}} = -2\tilde{V}_{ia}^{(\mathrm{xc})}$.

## C. Relevance of the Hessian

To first order in $\theta \in \mathbb{R}^{O \times V}$, the molecular orbital coefficients are given by

$$\mathbf{C}(\boldsymbol{\theta}; \mathbf{C}_0) = \mathbf{C}_0 + \mathbf{C}_0 \begin{pmatrix} 0 & \theta_{\mathrm{ov}} \\ -\theta_{\mathrm{ov}}^T & 0 \end{pmatrix} + \mathcal{O}(\boldsymbol{\theta}^2) \tag{31}$$

Expanding the energy in $\boldsymbol{\theta} \in \mathbb{R}^{O \times V}$ around the ground-state coefficients $C^*$

$$E_{\mathrm{tot}}(\boldsymbol{\theta}) = E_{\mathrm{tot}}^* + \underbrace{\frac{\partial E_{\mathrm{tot}}}{\partial \theta_{ia}}}_{=0} \theta_{ia} + \frac{1}{2} \frac{\partial^2 E_{\mathrm{tot}}}{\partial \theta_{ia} \partial \theta_{jb}} \theta_{ia} \theta_{jb} + \mathcal{O}(\boldsymbol{\theta}^3) \tag{32}$$

$$\frac{\partial^2 E_{\mathrm{xc}}}{\partial \theta_{ia} \partial \theta_{jb}} = \underbrace{2 \, C_{\mu a} \left. \frac{\partial V_{\mu\nu}^{\mathrm{xc}}}{\partial \theta_{jb}} \right|_0 C_{\nu i}}_{\text{kernel term}} + \underbrace{2 \left( \delta_{ij} \, C_{\mu a} V_{\mu\nu}^{\mathrm{xc}} C_{\nu b} - \delta_{ab} \, C_{\mu i} V_{\mu\nu}^{\mathrm{xc}} C_{\nu j} \right)}_{\text{potential terms}} \tag{33}$$

## D. Hessian-Vector Products on the Grassmannian.

Let $\boldsymbol{\theta} := \text{vec}(\theta_{ia}) \in \mathbb{R}^{OV}$ denote the vectorized (flattened) occupied-virtual orbital rotation angles, which parameterize the Grassmannian manifold $\text{Gr}(O, B)$. The orbital rotation Hessian

$$\boldsymbol{H} \in \mathbb{R}^{OV \times OV}, \qquad \boldsymbol{H}_{iajb} = \frac{\partial^2 E}{\partial \theta_{ia} \partial \theta_{jb}}\bigg|_{\boldsymbol{\theta}=0}$$

characterizes the local curvature of the energy surface at a stationary point. Explicitly, the Hessian-vector product with a tangent direction $\delta\boldsymbol{\theta} \in \mathbb{R}^{OV}$ reads:

$$\boldsymbol{H}\,\delta\boldsymbol{\theta} = \begin{bmatrix} \frac{\partial^2 E}{\partial\theta_1\partial\theta_1} & \frac{\partial^2 E}{\partial\theta_1\partial\theta_2} & \cdots & \frac{\partial^2 E}{\partial\theta_1\partial\theta_{OV}} \\ \frac{\partial^2 E}{\partial\theta_2\partial\theta_1} & \frac{\partial^2 E}{\partial\theta_2\partial\theta_2} & \cdots & \frac{\partial^2 E}{\partial\theta_2\partial\theta_{OV}} \\ \vdots & \vdots & \ddots & \vdots \\ \frac{\partial^2 E}{\partial\theta_{OV}\partial\theta_1} & \frac{\partial^2 E}{\partial\theta_{OV}\partial\theta_2} & \cdots & \frac{\partial^2 E}{\partial\theta_{OV}\partial\theta_{OV}} \end{bmatrix} \begin{bmatrix} \delta\theta_1 \\ \delta\theta_2 \\ \vdots \\ \delta\theta_{OV} \end{bmatrix} = \begin{bmatrix} \sum_{jb} \frac{\partial^2 E}{\partial\theta_1\partial\theta_{jb}}\,\delta\theta_{jb} \\ \sum_{jb} \frac{\partial^2 E}{\partial\theta_2\partial\theta_{jb}}\,\delta\theta_{jb} \\ \vdots \\ \sum_{jb} \frac{\partial^2 E}{\partial\theta_{OV}\partial\theta_{jb}}\,\delta\theta_{jb} \end{bmatrix}.$$

Choosing $\delta\boldsymbol{\theta} = \mathbf{e}_{jb}$ (a unit vector corresponding to a single occupied-virtual pair) extracts the $(jb)$-th column of the Hessian, which equals the linear response of the gradient:

$$\boldsymbol{H}\,\mathbf{e}_{jb} = \begin{bmatrix} \frac{\partial^2 E}{\partial\theta_1\partial\theta_{jb}} \\ \frac{\partial^2 E}{\partial\theta_2\partial\theta_{jb}} \\ \vdots \\ \frac{\partial^2 E}{\partial\theta_{OV}\partial\theta_{jb}} \end{bmatrix} = \frac{\partial \boldsymbol{g}^{(\text{xc})}}{\partial\theta_{jb}}\bigg|_{\boldsymbol{\theta}=0}.$$

For a general tangent vector $\delta\boldsymbol{\theta} \in T_P\text{Gr}(O, B)$, the result $\delta\boldsymbol{g}^{(\text{xc})} = \boldsymbol{H}\,\delta\boldsymbol{\theta}$ is the linear response of the XC gradient, with each component being a weighted sum over Hessian elements.

**Efficient Computation via Automatic Differentiation.** Materializing the full Hessian $\boldsymbol{H}$ requires $\mathcal{O}((OV)^2)$ storage and $\mathcal{O}((OV)^2)$ gradient evaluations, which becomes prohibitive for large systems. Instead, we leverage the fact that HVPs can be computed in $\mathcal{O}(OV)$ time and memory—the same cost as a single gradient evaluation—using forward-mode automatic differentiation.

The key identity is

$$\boldsymbol{H}\,\delta\boldsymbol{\theta} = \frac{\partial}{\partial\varepsilon}\bigg|_{\varepsilon=0} \boldsymbol{g}^{(\text{xc})}(\boldsymbol{\theta} + \varepsilon\,\delta\boldsymbol{\theta}),$$

i.e., the HVP is the Jacobian-vector product (JVP) of the gradient function. This operation can be vectorized (batches) over multiple directions, enabling the computation of $T$ independent HVPs with minimal overhead.

**Computational Complexity.**

| Operation | Time | Memory |
|---|---|---|
| Full Hessian $\boldsymbol{H}$ | $\mathcal{O}((OV)^2 \cdot C_g)$ | $\mathcal{O}((OV)^2)$ |
| Single HVP $\boldsymbol{H}\delta\boldsymbol{\theta}$ | $\mathcal{O}(C_g)$ | $\mathcal{O}(OV)$ |
| $T$ batched HVPs | $\mathcal{O}(T \cdot C_g)$ | $\mathcal{O}(T \cdot OV)$ |

Here $C_g$ denotes the cost of a single gradient evaluation. By supervising $T \ll OV$ randomly sampled linear responses, we obtain stochastic curvature information at a fraction of the cost of the full Hessian, making derivative-informed training tractable for systems with hundreds of basis functions. Similar HVP-based approaches for functional learning have been investigated in classical-DFT settings (Dijkman et al., 2025)

## E. Categorization of Traditional XC-Models: Jacob's Ladder

Perdew (2001) established a commonly referred to ranking of XC-models a.k.a. the *Jacob's Ladder*, with each rung representing an increase in expressivity, computational cost, and accuracy.

### E.1. Grid Features

Semi-local functionals depend on quantities evaluated at each point of a numerical integration grid:

- **Electron density:** $n = n_\uparrow + n_\downarrow$

- **Spin polarization:** $\zeta = (n_\uparrow - n_\downarrow)/n \in [-1, 1]$

- **Reduced density gradient:** $s = |\nabla n|/(2(3\pi^2)^{1/3} n^{4/3})$, measuring density inhomogeneity relative to the Fermi wavelength

- **Kinetic energy density:** $\tau = \frac{1}{2} \sum_{\mu\nu} P_{\mu\nu} \nabla \chi_\mu \cdot \nabla \chi_\nu$, encoding orbital structure

### E.2. Local and Semi-local Models

These functionals express the XC energy as $E_{\text{xc}}[n] = \int n(\mathbf{r})\, \varepsilon_{\text{xc}}(\mathbf{r})\, d\mathbf{r}$, differing in which grid features $\varepsilon_{\text{xc}}$ may depend on:

- **Local Density Approximation (LDA):** $\varepsilon_{\text{xc}}(n, \zeta)$

- **Generalized Gradient Approximation (GGA):** $\varepsilon_{\text{xc}}(n, \zeta, s)$

- **meta-GGA:** $\varepsilon_{\text{xc}}(n, \zeta, s, \tau)$

### E.3. Non-local Models

Higher rungs incorporate explicit orbital dependence, breaking the semi-local form.

**Hybrid functionals**   mix semi-local exchange with a fraction of exact (Hartree–Fock) exchange:

$$E_{\text{xc}}^{\text{hybrid}} = a\, E_x^{\text{HF}} + (1 - a)\, E_x^{\text{DFT}} + E_c^{\text{DFT}}. \tag{34}$$

**Range-separated hybrids**   partition the Coulomb operator $1/r_{12}$ into short- and long-range components using the error function, applying different exchange treatments to each range.

**Double hybrids**   additionally incorporate MP2-like correlation (Goerigk & Grimme, 2014):

$$E_c^{\text{PT2}} = \frac{1}{4} \sum_{ij}^{\text{occ}} \sum_{ab}^{\text{virt}} \frac{|\langle ij||ab\rangle|^2}{\varepsilon_i + \varepsilon_j - \varepsilon_a - \varepsilon_b}. \tag{35}$$

Range-separated variants replace the bare two-electron integrals with a range-separated interaction $\hat{g}_\omega(r)$:

$$E_c^{\text{PT2}}(\omega) = \frac{1}{4} \sum_{ij}^{\text{occ}} \sum_{ab}^{\text{virt}} \frac{|(ia|\hat{g}_\omega|jb)|^2}{\varepsilon_i + \varepsilon_j - \varepsilon_a - \varepsilon_b}. \tag{36}$$

## F. Time-Dependent DFT (TDDFT)

To find the resonance frequencies $\omega_n$ and thereby the excited state energies $E_n = E_0 + \omega_n$, we have to solve the Casida equations, a non-Hermitian eigenvalue problem given as

$$\begin{pmatrix} \mathbf{A} & \mathbf{B} \\ \mathbf{B}^* & \mathbf{A}^* \end{pmatrix} \begin{pmatrix} X_n \\ Y_n \end{pmatrix} = \omega_n \begin{pmatrix} \mathbf{1} & \mathbf{0} \\ \mathbf{0} & -\mathbf{1} \end{pmatrix} \begin{pmatrix} X_n \\ Y_n \end{pmatrix} \tag{37}$$

The eigenvectors $(X, Y)$ can be used to calculate properties like the transition amplitudes, intensity and dynamic polarizabilities. The submatrices $A$ and $B$ are constructed from orbital derivatives of the DFT functional:

$$A_{ia,jb} = \delta_{ij}\delta_{ab}(\varepsilon_a - \varepsilon_i) + K_{ia,jb} \tag{38}$$

$$K_{ia,jb} = \iint \psi_i(\mathbf{r})\psi_a(\mathbf{r}) \left[ \frac{1}{|\mathbf{r} - \mathbf{r}'|} + f_{xc}(\mathbf{r}, \mathbf{r}') \right] \psi_j(\mathbf{r}')\psi_b(\mathbf{r}') \, d\mathbf{r} \, d\mathbf{r}' \tag{39}$$

with the kernel $f_{xc}(\mathbf{r}, \mathbf{r}') = \frac{\delta^2 E_{xc}}{\delta\rho(\mathbf{r})\delta\rho(\mathbf{r}')}$, and

$$B_{ia,jb} = K_{ia,bj} \tag{40}$$

Since the Casida matrix is extremely large, full diagonalization is intractable. Instead, most implementations resort to iterative diagonalization with the Davidson algorithm. Our implementation follows the implementation given in (Olsen et al., 1988). TDDFT can have numerical convergence problems, which can often be improved on using the TDA approximation that sets $\mathbf{B} = 0$. As we can see from the above, TDDFT depends on the molecular energies, which in turn depend on the first functional derivative of the exchange-correlation functional, and the exchange-correlation kernel, which is the second functional derivative.

## G. The Optimized Effective Potential (OEP)

A hybrid exchange-correlation functional depends explicitly on the density matrix $P$, taking the form $E_{xc}[\rho, P]$. The minimization of this energy with respect to the orbitals naturally yields a non-local potential operator, $\Sigma_{xc}(\mathbf{r}, \mathbf{r}')$, defined by the functional derivative with respect to the density matrix:

$$\Sigma_{xc}(\mathbf{r}, \mathbf{r}') = \frac{\delta E_{xc}}{\delta P(\mathbf{r}, \mathbf{r}')} \tag{41}$$

In the context of pure Kohn-Sham DFT, however, the existence of a non-interacting system reproducing the interacting density relies on the Hohenberg-Kohn theorem, which requires a multiplicative, local potential $V_{xc}(\mathbf{r})$. A functional of the form $E_{xc}[\rho]$ yields such a local potential via the functional derivative with respect to the density:

$$V_{xc}(\mathbf{r}) = \frac{\delta E_{xc}}{\delta\rho(\mathbf{r})} \tag{42}$$

The fundamental challenge in constructing local approximations for orbital-dependent functionals lies in finding a local potential $V_{xc}^{\text{OEP}}(\mathbf{r})$ that best imitates the energetic effects of the true non-local operator $\Sigma_{xc}$.

**The OEP Condition:** To resolve this conflict within the rigorous framework of Kohn-Sham theory, the Optimized Effective Potential (OEP) method determines the local potential variationally. The optimal local potential is defined as the one for which the total energy, including the orbital-dependent terms, is stationary. Mathematically, this is formulated by the Sharp-Horton condition (which is mathematically equivalent to the Hohenberg-Kohn variational principle), which enforces that the first variation of the total energy with respect to the local potential vanishes:

$$\frac{\delta E_{tot}}{\delta V_{KS}(\mathbf{r})} = 0 \tag{43}$$

This condition ensures that the resulting local potential minimizes the energy within the constraints of a multiplicative Kohn-Sham operator.

Applying the chain rule to the Sharp-Horton condition and utilizing the response function of the non-interacting system leads to a Fredholm integral equation of the first kind (Ivanov et al., 2002):

$$\int \chi_s(\mathbf{r}, \mathbf{r}')\hat{V}_{xc}^{\text{OEP}}(\mathbf{r}')d\mathbf{r}' = \Lambda(\mathbf{r}) \tag{44}$$

where $\chi_s(\mathbf{r}, \mathbf{r}')$ is the static density-density response function of the non-interacting Kohn-Sham system:

$$\chi_s(\mathbf{r}, \mathbf{r}') = \sum_i^{\text{occ}} \sum_a^{\infty} \frac{\phi_i^*(\mathbf{r})\phi_a(\mathbf{r})\phi_a^*(\mathbf{r}')\phi_i(\mathbf{r}')}{\varepsilon_i - \varepsilon_a} + c.c. \tag{45}$$

(c.c. is the complex conjugate) and the inhomogeneity $\Lambda(\mathbf{r})$ collects the contributions arising from the non-local exchange-correlation operator:

$$\Lambda(\mathbf{r}) = \sum_i^{\text{occ}} \sum_a^{\infty} \frac{\langle a|\hat{\Sigma}_{xc}|i\rangle, \phi_i^*(\mathbf{r})\phi_a(\mathbf{r})}{\varepsilon_i - \varepsilon_a} + c.c. \tag{46}$$

It is not obvious that there exists a local $\hat{V}_{xc}^{\text{OEP}}$ that can solve equation (44). In fact, it is known that there can exist systems with densities, that cannot be reproduced by a non-interacting single slater determinant moving in a local potential $V_s(\mathbf{r})$ (Lieb, 1983) (mainly if the ground state is degenerate).

Even if the solution is $v_s$ representable, it is known that Fredholm Integral equations of the first kind can become hypersensitive, which makes their numerical solution ill-posed. Additionally, in practice, we have to use a finite orbital or auxiliary basis to discretize equation (44), which then reduces to a linear system of equations:

$$\sum_\nu \chi_{\mu\nu}, (\hat{V}_{xc}^{\text{OEP}})_\nu = \Lambda_\mu \tag{47}$$

The finite basis set expansion makes the kernel degenerate, and therefore unsolvable. To deal with these problems, usually only a regularized version of (47) is being solved (Ivanov et al., 2002).

In OEP KS-DFT, at every step of the SCF, we would have to take the current set of orbitals, solve (the regularized) equation (48) to get the new local $\hat{V}_{xc}^{\text{OEP}}$ to build our Hamiltonian matrix that we then diagonalize to get the new set of orbitals.

**Simplified OEP as a regularizer**  Given that solving equation (47) is very expensive and in practice not well posed to begin with, we do not use the OEP directly to supervise our model, but instead use a simpler regularizer merely inspired by the OEP. After some basic algebra, equation (44) can alternatively be expressed as a sum over occupied ($i$) and virtual ($a$) states (Kümmel & Kronik, 2008):

$$\sum_i^{\text{occ}} \sum_a^{\infty} \frac{\left(\langle a|\hat{V}_{xc}^{\text{OEP}}|i\rangle - \langle a|\hat{\Sigma}_{xc}|i\rangle\right)\phi_i^*(\mathbf{r})\phi_a(\mathbf{r})}{\varepsilon_i - \varepsilon_a} + c.c. = 0 \tag{48}$$

Here, $\varepsilon_i$ and $\varepsilon_a$ are the Kohn-Sham eigenvalues. The term $\langle a|\hat{\Sigma}_{xc}|i\rangle = \frac{dE_{xc}[P]}{dP}_{ai}$ represents the matrix element of the orbital-specific non-local operator, while $\langle a|\hat{V}_{xc}^{\text{OEP}}|i\rangle = \frac{dE_{xc}[\rho[P]]}{dP}_{ai}$ is the corresponding matrix element of the local potential. Formally, the sum over virtuals $a$ must run to infinity, though in practice it is truncated by the finite basis set size again.

Looking at equation (48) we see that the exact OEP equation requires only that the weighted sum of the differences between the local and non-local matrix elements vanishes at every point in space. A stricter condition is to demand that the matrix elements match individually for every occupied-virtual pair:

$$\langle a|\hat{V}_{xc}|i\rangle = \langle a|\hat{\Sigma}_{xc}|i\rangle \quad \forall i, a \tag{49}$$

If this condition were met, the numerator in the OEP equation would vanish term-by-term, trivially satisfying the integral equation. While the local potential generally lacks sufficient degrees of freedom to satisfy this equality strictly for all pairs, enforcing this condition in a least-squares sense (e.g. via DI-Loss) regularizes our functional and guides the optimization toward an OEP-like potential.

# H. Adaptive Training Stabilization

As explicitly noted by Luise et al. (2025), the online learning of the XC-functional in an SCF solver presents a challenge. During our evaluations, we noticed the same, due to the interaction between density optimization and parameter optimization. If the parameter update from the previous Self-Consistent Field (SCF) computation leads to an unstable XC-potential, which does not properly converge to a plausible density, the next parameter update becomes even worse. This type of training instability and architecture brittleness has been extensively explored in the literature on so-called *Deep Equilibrium models* (DEQs) (Bai et al., 2019; 2021; Agarwala & Schoenholz, 2022). By matching the established notation $z^* = f_\theta(x, z^*)$ of DEQs, this connection becomes immediate

$$P^* = \text{SCF}_\theta(P_0, P^*) = \text{SCF}(P_0, F_\theta(P_*)) . \tag{50}$$

We note that additional connections can be drawn for convergence acceleration techniques proposed in the context of DEQs and SCF calculations (Anderson, 1965; Pulay, 1980).

Training neural network XC functionals end-to-end through SCF requires safeguards against destabilizing parameter updates. We employ a Metropolis-inspired accept-reject mechanism that adaptively identifies outlier updates based on the relative change in mean epoch loss. The mechanism uses a tolerance schedule that relaxes constraints during early training, then tightens as the model stabilizes. Let $\tau_\text{w}$ denote the warmup length in epochs, and let $\Delta l_\text{tol}^{(0)}$ and $\Delta l_\text{tol}$ denote the initial and target tolerance thresholds, respectively. We define a piecewise linear tolerance schedule:

$$\Delta l_\text{tol}(t) = \begin{cases} \Delta l_\text{tol}^{(0)} & 0 \leq t < \frac{\tau_\text{w}}{2}, \\ \Delta l_\text{tol}^{(0)} + \dfrac{\Delta l_\text{tol} - \Delta l_\text{tol}^{(0)}}{\tau_\text{w}/2} \left(t - \frac{\tau_\text{w}}{2}\right) & \frac{\tau_\text{w}}{2} \leq t < \tau_\text{w}, \\ \Delta l_\text{tol} & t \geq \tau_\text{w}. \end{cases} \tag{51}$$

Early in training, the fixed-point map may only be marginally attractive, making SCF convergence fragile. We therefore start with a tighter tolerance $\Delta l_\text{tol}^{(0)}$ to aggressively reject destabilizing updates, then linearly relax to the target tolerance $\Delta l_\text{tol}$ over the second half of warmup as the model stabilizes, so as not to interfere excessively with the gradient-based optimizer.

At each epoch $t$, we compute the relative loss change $\Delta l_t = (L_t - L_{t-1})/L_{t-1}$ where $L_t$ is the mean epoch loss. We maintain an exponential moving average estimate of the *uncentered* standard deviation

$$\hat{\sigma}_t^2 = \beta \hat{\sigma}_{t-1}^2 + (1 - \beta)(\Delta l_t)^2, \tag{52}$$

which captures the typical scale of epoch-to-epoch loss fluctuations similar to the step-to-step variance estimate of the Adam optimizer (Kingma & Ba, 2017). To ensure numerical stability, we clip this estimate to a predefined range:

$$\tilde{\sigma}_t = \text{clip}(\hat{\sigma}_t, \sigma_\text{min}, \sigma_\text{max}). \tag{53}$$

The lower bound $\sigma_\text{min}$ prevents the mechanism from becoming overly sensitive to small fluctuations, which would artificially throttle training. Overly tight thresholds would erroneously reject updates important for informing the inner gradient-based optimizer. The **scaled deviation** $z_t$ measures how many standard deviations the current loss change exceeds the tolerance:

$$z_t = \frac{\Delta l_t}{\tilde{\sigma}_t}. \tag{54}$$

**Cosine decay function.**  To smoothly interpolate between full acceptance and full rejection, we define:

$$c(x; s) = \frac{1}{2} \left(1 + \cos\left(\pi \frac{\text{clip}(x, 0, s)}{s}\right)\right), \tag{55}$$

which transitions from 1 at $x = 0$ to 0 at $x = s$.

**Adaptive rejection factor.** Let $\kappa_t$ denote the tolerance in units of standard deviations (annealing from 1 to $\kappa$ over warmup) and $h_t = \kappa_t/2$ its half-width. The acceptance probability $p_t \in [0, 1]$ is:

$$p_t = \begin{cases} 1, & z_t < h_t, \\ c(z_t - h_t; h_t), & z_t \geq h_t. \end{cases} \tag{56}$$

Updates with $z_t < h_t$ are always accepted. For larger deviations, we accept the update with probability $p_t$, which decays smoothly via the cosine function, reaching zero (certain rejection) when $z_t \geq \kappa_t$. After consecutive rejections, we additionally rescale the optimizer momentum to prevent the accumulation of destabilizing gradient directions.

**Initialization selection.** The choice of initial network parameters has an outsized effect on end-to-end SCF training. Agarwala & Schoenholz (2022) show that DEQs are unusually sensitive to the statistics of their initialization, with poorly scaled weights yielding ill-conditioned or non-contractive fixed-point maps. Since our SCF map inherits this DEQ structure, an unfortunate initialization can place the XC functional in a regime where the iteration fails to reach a plausible density from the outset, before the stabilizer has any loss signal to act on. Rather than hand-tuning the initialization statistics, we treat the initialization as a cheap outer-loop meta-optimization: at the start of training we draw $n_{\text{init}}$ independent random parameter initializations, run a single epoch with each from a fresh optimizer state, and continue the main run from the parameters and optimizer state that achieved the lowest first-epoch loss. The selection is performed once and is negligible relative to the full training, while reliably discarding the small fraction of initializations that would otherwise stall SCF convergence. This concerns the network parameters only; the electron density is always initialized from a standard MINAO guess.

*Table 1.* Adaptive Training Stabilization

| Parameter | Value | Notes |
|---|---|---|
| Warm-up schedule | 10 epochs | First 10 epochs strict improvement, rejecting any updates that increase the loss |
| | 10 epochs | followed by a 10 epoch linear tolerance schedule until specified tolerance is reached |
| Tolerance | $5\,\sigma$ | Best out of $\{4\,\sigma, 5\,\sigma, 6\,\sigma\}$ |
| Lower bound on $\widehat{\sigma}^{(i)}$ | 20% | Best out of $\{15\%, 20\%, 25\%\}$ |
| Variance estimator momentum $\beta$ | 0.75 | Best out of $\{0.75, 0.8, 0.9\}$ differences are marginal |
| Parameter momentum rescaling | 0.7 | applied at each reject after 2 consecutive rejects (also tried 0.55 and 0.85) |
| Initialization tryouts $n_{\text{init}}$ | 5 | Random parameter inits trained for one epoch; lowest first-epoch loss retained |

# I. Optimizer Ablations

*Table 2.* Optimizer configuration for B3LYP/def2-SVP on QM9 (train up to 7 heavy atoms; test on 1000 molecules with exactly 9 heavy atoms) experiments

| Parameter | SCAN/def2-TZVPD | B3LYP/def2-SVP | Notes |
|---|---|---|---|
| Optimizer | muon | " | See head-to-head comparison to Adam in Table 3 |
| Warmup | linear | " | From EG-XC |
| Decay | $\frac{1}{1+i/N_{\text{scale}}}$ | " | Infinite range while monotonically decreasing |
| lr | 3e-2 / 1e-2 / 1e-1 | " | NNmGGA / XCdiff / Skala-mGGA best out of $\{$1e-1, 3e-2, 1e-2, 3e-3$\}$ |
| + warmup steps | 1000 | 1000 | From EG-XC |
| + decay scale | 750 | 1000 | EG-XC reference used 1000 throughout, we observed quicker training with 750 for small subsplits of QM9 |
| Weight decay | 1e-5 | " | |

# J. Evaluation metrics

Here we list and describe all the metrics we are using:

*Table 3.* NNmGGA models on QM9, trained on molecules with up to 5 heavy atoms and evaluated on 1000 randomly selected molecules with exactly 9 heavy atoms, fixed across all runs, at SCAN/def2-TZVPD level of theory. Energy MAEs in m$E_h$. These are early-stage ablations with different hyperparameters than those we later settled on; nevertheless, the results are representative of the optimizer's performance. We observe higher training stability and improvements on almost all metrics across all loss compositions when using Muon (Jordan et al., 2024) compared to Adam (Kingma & Ba, 2017).

| Model | Loss | $E_{\text{tot}}$ | $E_\rho$ | $\mu_\rho$ | $E_C$ | $E_{\text{xc}}$ | $\Delta\varepsilon_{\text{HL}}$ | RC | RIC |
|---|---|---|---|---|---|---|---|---|---|
| Adam | $E_{\text{tot}}$ | $2.88 \pm .31$ | $59.81 \pm 8.93$ | $3.0 \times 10^{-2}$ | $130 \pm 50$ | $60.22 \pm 8.31$ | $22.56 \pm 0.73$ | $98.5\%$ | $1.02$ |
| | $+E_\rho$ | $2.77 \pm .32$ | $2.51 \pm 0.19$ | $3.5 \times 10^{-2}$ | $288 \pm 40$ | $3.90 \pm 0.48$ | $15.39 \pm 4.09$ | $100\%$ | $1.00$ |
| | $+\nabla$ | $1.16 \pm .18$ | $0.41 \pm 0.11$ | $1.8 \times 10^{-3}$ | $3.11 \pm .68$ | $1.27 \pm 0.06$ | $0.39 \pm 0.12$ | $100\%$ | $0.99$ |
| | $+H$ | $2.15 \pm 1.1$ | $0.48 \pm 0.10$ | $1.4 \times 10^{-3}$ | $4.47 \pm 2.2$ | $2.24 \pm 0.97$ | $0.31 \pm 0.07$ | $100\%$ | $0.99$ |
| Muon | $E_{\text{tot}}$ | $2.09 \pm .21$ | $58.48 \pm 6.96$ | $3.3 \times 10^{-2}$ | $138 \pm 10$ | $59.59 \pm 6.64$ | $21.36 \pm 1.91$ | $100\%$ | $1.04$ |
| | $+E_\rho$ | $2.44 \pm .72$ | $1.64 \pm 0.31$ | $3.3 \times 10^{-2}$ | $468 \pm 122$ | $3.08 \pm 1.34$ | $14.84 \pm 4.67$ | $99.7\%$ | $1.05$ |
| | $+\nabla$ | $0.38 \pm .14$ | $0.14 \pm 0.04$ | $9.0 \times 10^{-4}$ | $1.14 \pm .39$ | $0.34 \pm 0.13$ | $0.16 \pm 0.02$ | $100\%$ | $0.99$ |
| | $+H$ | $0.98 \pm .42$ | $0.14 \pm 0.03$ | $6.3 \times 10^{-4}$ | $1.20 \pm .29$ | $0.92 \pm 0.39$ | $0.11 \pm 0.04$ | $100\%$ | $1.00$ |

**Total energy $E_{\text{tot}}$:**   The total energy error between the self-consistent density of the learned functional $\rho_{\text{ML}}$ and the self-consistent reference density $\rho_{\text{ref}}$

$$\Delta E_{\text{tot}} = |E_{\text{tot,ML}}[\rho_{\text{ML}}] - E_{\text{tot,ref}}[\rho_{\text{ref}}]| \tag{57}$$

**Total energy error with donated density $E_{\text{ref}}$**   The energy error for the self-consistent density of the learned functional evaluated with the reference functional

$$\Delta E_{\text{ref}} = |E_{\text{tot,ref}}[\rho_{\text{ML}}] - E_{\text{tot,ref}}[\rho_{\text{ref}}]| \tag{58}$$

**Exchange energy $E_{\text{xc}}$:**   The XC energy error between the learned XC functional with its self-consistent density, and the reference XC functional with its own self-consistent reference density

$$\Delta E_{\text{xc}} = |E_{\text{xc},1}[\rho_{\text{ML}}] - E_{\text{xc,ref}}[\rho_{\text{ref}}]| \tag{59}$$

**Coulomb (Hartree) metric:**   The error of the Coulomb energy between the learned self-consistent density and the ground truth self-consistent density

$$\Delta E_C = |E_C[\rho_{\text{ML}}] - E_C[\rho_{\text{ref}}]| \tag{60}$$

**Mean-field energy error:**   The mean-field energy difference is defined as all the electronic energy contributions except for $E_{\text{xc}}$ (Gould, 2023)

$$F_{\text{MF}}[P] = T_s[P] + \int v_{\text{ext}}(\mathbf{r})\, \rho[P](\mathbf{r})\, d\mathbf{r} + E_{\text{H}}[P],$$

the corresponding density error is $|F_{\text{MF}}[P_{\text{ML}}] - F_{\text{MF}}[P_{\text{ref}}]|$ with the density matrix $P$.

**Homo-Lumo Gap $\Delta\varepsilon_{\text{HL}}$:**   The KS Homo-Lumo gap for a density $\rho$ is defined as

$$\Delta_{\text{HL}} = |\varepsilon_{\text{L}}[\rho_{\text{ML}}] - \varepsilon_{\text{H}}[\rho_{\text{ref}}]|,$$

where $\varepsilon_{\text{H}}$ and $\varepsilon_{\text{L}}$ are the highest occupied and lowest unoccupied KS eigenvalues of the effective one-electron Hamiltonian corresponding to $\rho[P]$. Given a reference density $\rho_{\text{ref}}$, the Homo-Lumo gap error is then

$$\Delta\varepsilon_{\text{HL}} = |\Delta_{\text{HL}}[\rho[P_{\text{ML}}]] - \Delta_{\text{HL}}[\rho[P_{\text{ref}}]].|$$

$L_1$ **norm:**   The $L_1$ distance

$$L_1[\Delta\rho] = \int |\rho_{\text{ML}}(\mathbf{r}) - \rho_{\text{ref}}(\mathbf{r})|\, d\mathbf{r},$$

uniformly measures the total absolute difference between two densities.

$L_2$ **norm:**   The $L_2$ distance

$$L_2[\Delta\rho] = \left( \int |\rho_{\mathrm{ML}}(\mathbf{r}) - \rho_{\mathrm{ref}}(\mathbf{r})|^2 \, d\mathbf{r} \right)^{1/2},$$

emphasizes larger local deviations more strongly than the $L_1$ norm and is therefore more sensitive to large spikes that can occur, for example, around the core.

**Dipole error:**   The dipole difference,

$$\boldsymbol{\mu}_{\Delta\rho} = |\mu[\rho_{\mathrm{ML}}] - \mu[\rho_{\mathrm{ref}}]|,$$

captures how discrepancies in the density translate into errors in the first moment of the charge distribution. It probes global shifts of electronic charge and is directly related to experimentally observable response properties.

**Excited state energies:**   We run TDDFT with the learned and reference functional at their respective self-consistent densities to obtain the first 5 excited states and report their error.

## K. Runtime Benchmarks

To complement the possible runtime speedup analysis in Figure 3, we benchmark the walltime of a single SCF cycle for B3LYP and the distilled NNmGGA functional. Each timing is averaged over ten repetitions on one NVIDIA H200 GPU. All measurements are from our unoptimized research codebase and should be interpreted as indicative single-cycle costs rather than production-ready inference timings. We use PySCF grid level 1 throughout these experiments. This grid choice reduces peak memory consumption, since our current implementation keeps the density-fitted ERI tensor in GPU memory rather than rematerializing it on the fly as in production-ready inference-only DFT codes.

*Table 4.* Single-cycle SCF walltime benchmarks on QM9 molecules. Timings are reported in milliseconds as mean $\pm$ standard deviation over ten repetitions on one NVIDIA H200 GPU. The speedup is computed as the B3LYP walltime divided by the NNmGGA walltime.

| Basis | # Heavy | # Atoms | # Basis | # Grid | B3LYP | NNmGGA | Speedup |
|---|---|---|---|---|---|---|---|
| def2-SVP | 10 | 30 | 240 | 101240 | $19.0 \pm 0.3$ | $17.3 \pm 0.1$ | $1.10\times$ |
| | 19 | 42 | 381 | 155216 | $94.3 \pm 1.5$ | $53.4 \pm 0.1$ | $1.77\times$ |
| | 30 | 58 | 560 | 224376 | $294.1 \pm 11.0$ | $113.5 \pm 1.6$ | $2.59\times$ |
| | 37 | 67 | 668 | 265340 | $596.4 \pm 11.5$ | $186.9 \pm 2.5$ | $3.19\times$ |
| def2-TZVPD | 10 | 30 | 550 | 101240 | $150.7 \pm 5.8$ | $56.3 \pm 0.7$ | $2.67\times$ |
| | 19 | 42 | 916 | 155216 | $1004.2 \pm 11.0$ | $182.0 \pm 3.6$ | $5.52\times$ |

The NNmGGA pre-run includes neural network evaluation and is already comparable to B3LYP for small def2-SVP molecules. The speedup increases consistently with system size and basis size, reaching more than $5\times$ at the def2-TZVPD level for 19 heavy atoms. This trend reflects the favorable $\mathcal{O}(B^3)$ scaling of semi-local ML-XC functionals relative to the $\mathcal{O}(B^4)$ exact-exchange contribution in B3LYP. At the smallest system sizes the neural network evaluation introduces a modest overhead, but this overhead is quickly amortized as exact exchange dominates the hybrid-functional cost.

For a fully optimized, production-ready integration of a related architecture, Luise et al. (2025) report GPU-accelerated Skala runtimes matching semi-local r2SCAN cost and more than $10\times$ below hybrid-functional cost. Their CPU-based PySCF implementation shows an approximately 3–4$\times$ prefactor over r2SCAN, consistent with the overhead observed in our unoptimized implementation.

### K.1. End-to-End Initialization Runtime

We additionally measure end-to-end walltime for using an ML-XC pre-run to initialize B3LYP on a randomly selected large QM40 molecule with 35 heavy atoms at the B3LYP/def2-SVP level of theory. All workflows converge to the same final B3LYP energy of $-1587.69540087 \, E_h$. The baseline is a direct B3LYP calculation from the standard initialization, while the remaining workflows first perform a fixed number of NNmGGA SCF cycles and then continue with B3LYP from the resulting density.

*Table 5.* End-to-end walltime for ML-XC initialized B3LYP calculations on a 35-heavy-atom QM40 molecule at the B3LYP/def2-SVP level of theory. Walltimes are reported in seconds as mean ± standard deviation. The speedup is relative to direct B3LYP.

| Workflow | ML-XC Cycles | B3LYP Cycles | Total Cycles | Walltime | Speedup |
|---|---|---|---|---|---|
| Direct B3LYP | 0 | 14 | 14 | $4.969 \pm 0.007$ | $1.00\times$ |
| ML-XC + B3LYP | 4 | 11 | 15 | $4.422 \pm 0.008$ | $1.12\times$ |
| | 5 | 11 | 16 | $4.553 \pm 0.005$ | $1.09\times$ |
| | 6 | 8 | 14 | $3.673 \pm 0.008$ | $\mathbf{1.35\times}$ |
| | 7 | 8 | 15 | $3.784 \pm 0.006$ | $1.31\times$ |
| | 8 | 8 | 16 | $3.899 \pm 0.005$ | $1.27\times$ |
| | 9 | 8 | 17 | $4.012 \pm 0.006$ | $1.24\times$ |
| | 10 | 8 | 18 | $4.123 \pm 0.006$ | $1.21\times$ |
| | 11 | 8 | 19 | $4.244 \pm 0.006$ | $1.17\times$ |
| | 12 | 8 | 20 | $4.355 \pm 0.001$ | $1.14\times$ |
| | 13 | 8 | 21 | $4.472 \pm 0.008$ | $1.11\times$ |
| | 14 | 8 | 22 | $4.586 \pm 0.005$ | $1.08\times$ |

For this molecule, the best trade-off is reached after six NNmGGA cycles. This yields a net $1.35\times$ walltime speedup while keeping the total number of SCF cycles equal to the direct B3LYP calculation. Since the single-cycle benchmarks show increasing ML-XC speedups with system size, we expect the net end-to-end speedup to improve further for larger molecules.

### K.2. Training Wall-Clock Runtime

We also report training wall-clock curves for the B3LYP runs. All runs were performed on the same compute node with one H100 GPU used per run. Figure 5 tracks training progress and validation energy error as a function of wall time for each loss variant. All runs use the same early-stopping parameters, and most training sessions finish at comparable wall-clock times. The per-step overhead of Hessian supervision is visible as fewer completed optimization steps at a fixed wall time, but the total training duration remains comparable across loss variants. On a single H100 GPU, the approximate per-step times are 300 ms for the $E + \rho$ density baseline, 310 ms with gradient supervision, and 400 ms with Hessian supervision, corresponding to an approximately 33% overhead for the Hessian term. The validation set is drawn from the same QM7 distribution as the training data; we therefore use validation energy error here only as a common, physically interpretable observable for tracking training progress over time.

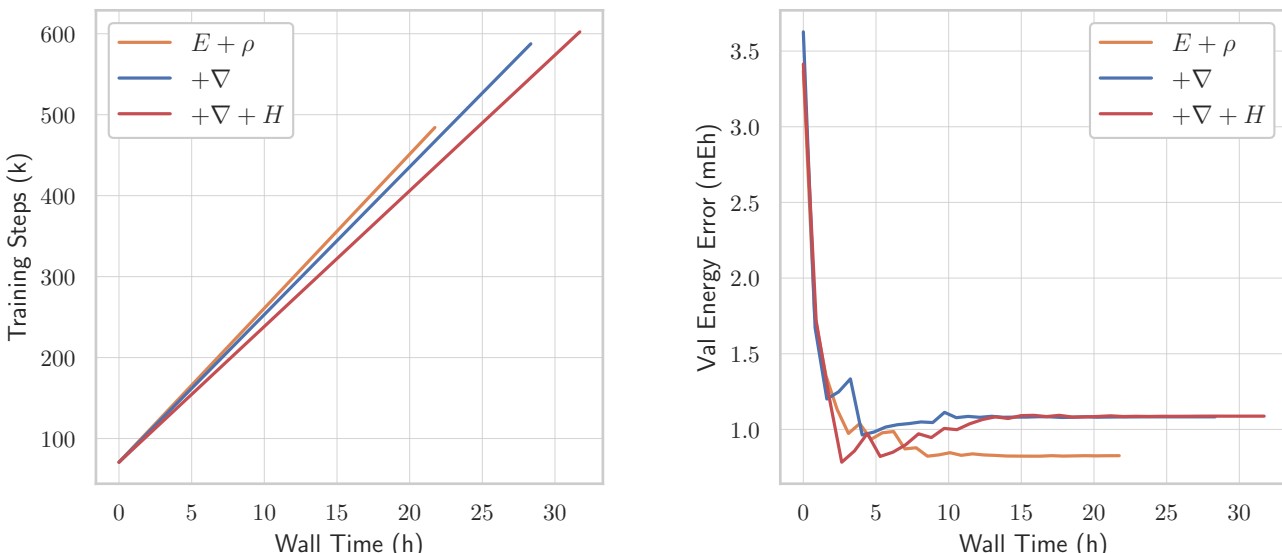

*Figure 5.* Training wall-clock curves for B3LYP density-baseline runs on a single H100 GPU. The left panel shows training progress as a function of wall time, where Hessian supervision completes fewer steps at a fixed time because of its per-step overhead. The right panel shows validation energy error on the in-distribution QM7 validation split, used here as a common observable for comparing training progress.

# L. Data Tables

## L.1. Learning SCAN

*Table 6.* Local models on QM9, trained on molecules with up to 5 heavy atoms and evaluated on 1000 randomly selected molecules with exactly 9 heavy atoms, fixed across all runs, at SCAN/def2-TZVPD level of theory. Energy MAEs in $\text{mE}_h$.

| Model | Loss | $E_{\text{tot}}$ | $E_\rho$ | $\mu_\rho$ | $E_C$ | $E_{\text{xc}}$ | $\Delta\varepsilon_{\text{HL}}$ | RC | RIC |
|---|---|---|---|---|---|---|---|---|---|
| NNmGGA | $E_{\text{tot}}$ | $2.09 \pm .21$ | $58.48 \pm 6.96$ | $3.3 \times 10^{-2}$ | $138 \pm 10$ | $59.59 \pm 6.64$ | $21.36 \pm 1.91$ | 100% | 1.04 |
| | $+E_\rho$ | $2.44 \pm .72$ | $1.64 \pm 0.31$ | $3.3 \times 10^{-2}$ | $468 \pm 122$ | $3.08 \pm 1.34$ | $14.84 \pm 4.67$ | 99.7% | 1.05 |
| | $+\nabla$ | $0.38 \pm .14$ | $0.14 \pm 0.04$ | $9.0 \times 10^{-4}$ | $1.14 \pm .39$ | $0.34 \pm 0.13$ | $0.16 \pm 0.02$ | 100% | 0.99 |
| | $+H$ | $0.98 \pm .42$ | $0.14 \pm 0.03$ | $6.3 \times 10^{-4}$ | $1.20 \pm .29$ | $0.92 \pm 0.39$ | $0.11 \pm 0.04$ | 100% | 1.00 |
| XCdiff | $E_{\text{tot}}$ | $1.16 \pm .19$ | $64.95 \pm 16.9$ | $2.3 \times 10^{-2}$ | $135 \pm 72$ | $65.59 \pm 16.8$ | $16.69 \pm 5.65$ | 100% | 1.02 |
| | $+E_\rho$ | $1.40 \pm .47$ | $1.54 \pm 0.55$ | $2.1 \times 10^{-2}$ | $561 \pm 142$ | $2.57 \pm 0.51$ | $14.85 \pm 1.74$ | 99.5% | 1.09 |
| | $+\nabla$ | $0.22 \pm .01$ | $0.07 \pm 0.02$ | $7.1 \times 10^{-4}$ | $1.02 \pm .06$ | $0.21 \pm 0.00$ | $0.28 \pm 0.07$ | 100% | 1.00 |
| | $+H$ | $0.35 \pm .05$ | $0.05 \pm 0.01$ | $3.8 \times 10^{-4}$ | $0.68 \pm .10$ | $0.33 \pm 0.03$ | $0.20 \pm 0.04$ | 100% | 1.00 |
| Skala (mGGA) | $E_{\text{tot}}$ | $1.90 \pm .32$ | $33.71 \pm 4.49$ | $2.6 \times 10^{-2}$ | $198 \pm 60$ | $35.03 \pm 4.50$ | $11.92 \pm 1.87$ | 100% | 1.02 |
| | $+E_\rho$ | $1.28 \pm .10$ | $1.10 \pm 0.17$ | $2.8 \times 10^{-2}$ | $205 \pm 46$ | $1.46 \pm 0.12$ | $12.27 \pm 1.58$ | 99.8% | 1.02 |
| | $+\nabla$ | $1.40 \pm .59$ | $0.40 \pm 0.12$ | $3.4 \times 10^{-3}$ | $4.01 \pm 1.49$ | $1.41 \pm 0.62$ | $0.89 \pm 0.23$ | 99.9% | 1.00 |
| | $+H$ | $1.84 \pm .61$ | $0.35 \pm 0.08$ | $2.8 \times 10^{-3}$ | $3.08 \pm 0.52$ | $1.84 \pm 0.62$ | $0.39 \pm 0.04$ | 100% | 0.99 |

## L.2. Distilling B3LYP into Semi-Local ML-XC-Functionals

*Table 7.* Raw data of all late-stage runs performed for B3LYP/def2-SVP distillation across loss weight configurations. Each row corresponds to a different weighting of energy ($E$), density ($\rho$), gradient ($\nabla$), and Hessian ($H$) supervision terms. Results are reported for NNmGGA (Nagai et al., 2020), XCdiff (Dick & Fernandez-Serra, 2021), Skala-mGGA (Luise et al., 2025), EG-XC (Gao et al., 2024) trained on QM9 up to 7 heavy atoms and evaluated on a random subset of QM40. Best per model metrics are highlighted in bold. One molecule (ID 86802) has been removed from the QM40 evaluation due to convergence issues across all models which dominated the MAE, the affected rows are marked with an asterisk in the $N_{\text{Runs}}$ column. The grey rows are not plotted in the figures, but they informed the loss-weights chosen for the mGGA models (1e-3 for $\alpha_\nabla$ and 3e-5 for $\alpha_H$).

| MODEL | $N_{\text{RUNS}}$ | $\alpha_\nabla$ | $\alpha_H$ | $E_{\text{TOT}}$ | | $E_\rho$ | | $E_{\text{xc}}$ | | $\Delta\varepsilon_{\text{HL}}$ | | $\mu_\rho$ | $\mathcal{L}_2[\rho]$ | RIC | $E_{\text{B3LYP}}$ |
|---|---|---|---|---|---|---|---|---|---|---|---|---|---|---|---|
| NNMGGA | 4* | - | - | 4.2 | ±0.6 | 1.4 | ±0.2 | 3.0 | ±0.4 | 51.5 | ±0.1 | **0.0180** | **1.0E-5** | 1.16 | 0.9 |
| | 3* | 1E-3 | - | 5.7 | ±0.4 | 0.9 | ±0.1 | 5.4 | ±0.3 | 53.4 | ±0.3 | 0.0197 | 1.2E-5 | 1.17 | **0.7** |
| | 3* | 1E-3 | 1E-5 | 3.2 | ±0.3 | **0.8** | ±0.1 | 3.1 | ±0.2 | 50.9 | ±0.0 | 0.0190 | 1.2E-5 | 1.15 | 0.7 |
| | 3* | 1E-3 | 3E-5 | 2.9 | ±0.5 | 0.9 | ±0.2 | 2.6 | ±0.4 | 49.6 | ±0.1 | 0.0189 | 1.2E-5 | 1.15 | 0.7 |
| | 1 | 1E-3 | 1E-4 | **2.2** | - | 0.8 | - | **2.3** | - | **48.9** | - | 0.0205 | 1.3E-5 | **1.14** | 0.7 |
| SKALA_MGGA | 3* | - | - | 15.8 | ±1.8 | 1.2 | ±0.1 | 14.6 | ±1.8 | 50.3 | ±0.2 | **0.0176** | **9.1E-6** | 1.15 | 0.9 |
| | 3* | 1E-3 | - | 5.8 | ±1.5 | 0.9 | ±0.2 | 5.4 | ±1.3 | 52.6 | ±0.2 | 0.0192 | 1.3E-5 | 1.16 | 0.7 |
| | 2* | 1E-3 | 1E-5 | 4.7 | ±1.2 | 1.0 | ±0.2 | 4.2 | ±0.7 | 50.4 | ±0.1 | 0.0181 | 1.1E-5 | 1.15 | 0.7 |
| | 3* | 1E-3 | 3E-5 | **3.6** | ±0.2 | **0.7** | ±0.0 | **3.7** | ±0.2 | 49.5 | ±0.1 | 0.0182 | 1.1E-5 | **1.14** | **0.7** |
| XCDIFF | 3* | - | - | 8.9 | ±0.9 | 1.2 | ±0.1 | 7.9 | ±1.0 | 51.8 | ±0.1 | **0.0182** | 9.8E-6 | 1.17 | 0.9 |
| | 1 | 1E-4 | - | 11.7 | - | 1.0 | - | 10.9 | - | 52.0 | - | 0.0183 | **9.3E-6** | 1.17 | 0.8 |
| | 3* | 1E-3 | - | 3.7 | ±0.4 | 1.0 | ±0.0 | 3.4 | ±0.2 | 53.2 | ±0.1 | 0.0191 | 1.2E-5 | 1.18 | 0.7 |
| | 1* | 3E-3 | - | **2.2** | - | 1.1 | - | **2.6** | - | 53.8 | - | 0.0201 | 1.5E-5 | 1.19 | **0.6** |
| | 2* | 1E-3 | 1E-5 | 4.6 | ±0.6 | 0.8 | ±0.1 | 5.0 | ±0.4 | 50.5 | ±0.0 | 0.0182 | 1.1E-5 | 1.16 | 0.7 |
| | 3* | 1E-3 | 3E-5 | 5.5 | ±1.6 | **0.7** | ±0.0 | 5.5 | ±1.6 | 49.4 | ±0.0 | 0.0187 | 1.1E-5 | **1.14** | 0.7 |
| | 1* | 1E-3 | 1E-4 | 6.7 | - | 0.7 | - | 7.0 | - | **49.0** | - | 0.0195 | 1.2E-5 | 1.15 | 0.7 |
| EG-XC | 1* | - | - | 15.9 | - | **1.0** | - | 15.6 | - | 51.8 | - | 0.0182 | 1.0E-5 | **1.16** | 0.9 |
| | 1* | 1E-6 | - | 3.4 | - | 1.2 | - | **3.3** | - | 52.3 | - | **0.0179** | 9.8E-6 | 1.18 | **0.7** |
| | 1 | 1E-6 | 1E-5 | **3.1** | - | 1.0 | - | 3.4 | - | **49.2** | - | 0.0181 | **9.2E-6** | 1.16 | 0.7 |

# M. Machine Learnable XC-Models

## M.1. NN-mGGA

In their foundational work on machine-learned exchange-correlation functionals Nagai et al. (2020) circumvented the parameter gradient calculation by using a probabilistic (Metropolis-Hastings-type) training method. To make their model more amenable to gradient-based training methods, we replaced their $C^1$ activation function (ELU) by a $C^\infty$ activation (shifted softplus), while maintaining the same value range $\sigma : \mathbb{R} \to (-1, \infty)$ and zero fixed point $\sigma(0) = 0$. Moreover, we standardize the output space of the neural network by adding a scaling factor $s = 0.1$ and by initializing the bias of the output layer, s.t. $\sigma(b) = 0.1$ to account for the systematic bias of the uniform electron gas (UEG/LDA) on the stable organic chemistry datasets that we focus on in this work

$$F_\theta(f_\rho(r)) = 1 + \sigma(s \cdot \text{NN}_\theta(f_\rho(r)) + b) . \tag{61}$$

While this shift can be unlearned by the bias of the last layer, by shifting the output at initialization, the training already starts at a better point in parameter space. We verified all of these tweaks to aid model performance during late-stage development on QM9 molecules with up to 4 heavy atoms.

## M.2. Hyperparameters

Semi-local model sizes are matched to XCdiff within each teacher-functional setting to keep the architecture comparison focused on the effect of the loss terms. The larger B3LYP models reflect the higher capacity required to distill the non-local exact-exchange contribution of the hybrid reference.

*Table 8.* Semi-local model hyperparameters

| SCAN | # PARAMETERS | HIDDEN DIM. | DEPTH |
|---|---|---|---|
| NN-mGGA | 1243 | 23 | 4 |
| XCdiff | 1250 | 16 | 4 |
| Skala-mGGA | 1212 | 22 | 4 |
| **B3LYP** | | | |
| NN-mGGA | 4417 | 32 | 6 |
| XCdiff | 4648 | 23 | 6 |
| Skala-mGGA | 4513 | 32 | 6 |

*Table 9.* EG-XC specific hyperparameters (Gao et al., 2024)

| **Embedding** | | |
|---|---|---|
| Radial cutoff | 5.0 Å | From EG-XC |
| # radial filters | 32 | From EG-XC |
| nuclei partitioning | 'exponential' | From EG-XC |
| **GNN** | | |
| Hidden irreps | $128x0e + 64x1o$ | We deviate from the original $32x0e+32x1o+32x2e$ to improve compile speed and reduce memory requirements, while observing little to no performance loss |
| Radial basis size | 8 | From EG-XC |
| Message distance cutoff | 5.0 Å | From EG-XC |
| # layers | 3 | From EG-XC |
| **Non-Local Reweighting-MLP** | | |
| # non-local grid features | 16 | From EG-XC |
| Hidden dim. | 16 | From EG-XC |
| # layers | 3 | From EG-XC |
| Activation | SiLU | Infinitely smooth differentiable (from EG-XC) |
| initial output scale | 0.1 | Reduced from 1 for increased training stability |

## N. Density Loss-Type Ablations

*Table 10.* Density loss ablations with NNmGGA at B3LYP/def2-SVP level of theory on QM9, trained on molecules with up to 7 heavy atoms and evaluated on 1000 randomly selected molecules with exactly 9 heavy atoms, fixed across all runs (same setting as Tab. 6)

|  | $\log_{10} \alpha_\rho$ | $E_{\text{TOT}}$ | $E_{\text{XC}}$ | $\Delta\varepsilon_{\text{HL}}$ | $E_\rho$ | $\mu_\rho$ | $\mathcal{L}_1[\rho]$ | $\mathcal{L}_2[\rho]$ | $E_C$ |
|---|---|---|---|---|---|---|---|---|---|
| $E_\rho$ | $\pm 0$ | 2.02 | 1.67 | 53.95 | 1.46 | 0.0302 | 0.236 | 0.000420 | 179.28 |
| $L_1$ | $-4$ | 1.05 | 0.87 | 54.27 | 0.83 | 0.0113 | 0.0246 | $4 \times 10^{-6}$ | 7.32 |
|  | $-3$ | 0.97 | 1.12 | 53.71 | 0.81 | 0.0098 | 0.0210 | $3 \times 10^{-6}$ | 6.77 |
|  | $-2$ | 1.65 | 1.75 | 54.23 | 0.74 | 0.0103 | 0.0210 | $3 \times 10^{-6}$ | 6.59 |
|  | $-1$ | 5.13 | 4.97 | 54.70 | 0.86 | 0.0105 | 0.0222 | $4 \times 10^{-6}$ | 7.46 |
| $L_2$ | $-1$ | 0.80 | 1.02 | 56.09 | 0.54 | 0.0130 | 0.0291 | $4 \times 10^{-6}$ | 12.94 |
|  | $\pm 0$ | 1.21 | 1.27 | 54.99 | 0.47 | 0.0125 | 0.0261 | $3 \times 10^{-6}$ | 10.90 |
|  | $+1$ | 1.67 | 1.61 | 55.77 | 0.44 | 0.0127 | 0.0261 | $3 \times 10^{-6}$ | 10.52 |

We compare three choices for the density term in the $E + \rho$ baseline, the per-electron $L_1$ and $L_2$ norms of the real-space density error and the mean-field energy error $E_\rho$ (Gould, 2023). We sweep the weight $\alpha_\rho$ for each and report the resulting metrics in Table 10. The mean-field energy error gives poor real-space density agreement, which motivates the two real-space norms. The $L_1$ norm at $\alpha_\rho = 10^{-3}$ reaches the lowest dipole and $L_1[\rho]$ error, so we adopt it as the density term throughout. The $L_2$ norm reaches lower total and mean-field energy errors but worse density shape, which we do not prioritize since the energy term already supervises the energy.

