# OpenReview forum: "Derivative Informed Learning of Exchange-Correlation Functionals"
_ICML.cc/2026/Conference — ICML 2026 regular_

### Official Review · Reviewer_EnkS · 2026-02-28

**Soundness:** 3
**Presentation:** 3
**Significance:** 4
**Originality:** 3
**Overall Recommendation:** 5
**Confidence:** 2

**Summary:**

The paper proposes "Derivative Informed XC-Loss" (DI-Loss), a training objective for machine-learned Exchange-Correlation (XC) functionals in Density Functional Theory (DFT). \textbf{This paper's fundamental contribution pertains to} the regularization of the learning process by supervising not just the energy and density, but also the first and second derivatives (gradients and Hessians) of the energy with respect to the density matrix on the Grassmannian manifold. The authors utilize Hessian-Vector Products (HVPs) to efficiently compute these terms without materializing the full Hessian. The method is primarily evaluated in a "distillation" setting, where hybrid functionals (like B3LYP) are compressed into semi-local neural network functionals (like NNmGGA). The results demonstrate improved SCF convergence stability, better density properties, and reduced errors in excited-state predictions (TDDFT) compared to baselines trained only on energy and density.

**Compliance With Llm Reviewing Policy:**

Affirmed.

**Key Questions For Authors:**

\section*{Questions for Authors}
\begin{enumerate}
    \item \textbf{Scalability to Wavefunction Data:} The current method shines in distillation because the teacher (B3LYP) provides cheap gradients/Hessians. How do you propose to apply DI-Loss when training on high-accuracy wavefunction data (e.g., CCSD(T)) where the "true" Hessian is not readily available? Is the method strictly limited to functional distillation?
    \item \textbf{Hyperparameter Robustness:} How sensitive is the final performance to the ratio of $\alpha_\nabla / \alpha_H$? Is there a universal heuristic, or must this be re-tuned for every new architecture/dataset pair?
    \item \textbf{Comparison to Potential Regularization:} How does supervising the Hessian compare to directly supervising the XC potential ($v_{xc}$) via the OEP equation or density-inversion schemes? Is the second-order information strictly necessary, or would first-order potential matching suffice?
    \item \textbf{Computational Cost of Training:} What is the wall-clock time overhead of training with DI-Loss compared to the standard $E + \rho$ loss? The paper mentions the complexity class, but practical training times would be helpful.
\end{enumerate}

**Limitations:**

Yes

**Strengths And Weaknesses:**

\section*{Strengths}
\begin{itemize}
    \item \textbf{Physically Motivated Regularization:} The inclusion of gradient and Hessian information is theoretically sound. By enforcing the curvature of the energy landscape to match the reference, the method ensures that the learned functional mimics the dynamics of the reference functional, not just its fixed point.
    \item \textbf{Technical Implementation:} The use of Hessian-Vector Products (HVPs) via automatic differentiation to avoid the $O((OV)^2)$ cost of the full Hessian is a necessary and well-executed technical detail, making the method feasible for the system sizes considered.
    \item \textbf{Comprehensive Evaluation:} The authors evaluate the method across a wide range of metrics, including SCF convergence stability (RIC), dipole moments, and TDDFT excitation energies, rather than relying solely on ground-state energy errors.
    \item \textbf{Distillation Performance:} The empirical results show that the method successfully distills the accuracy of hybrid functionals into lower-cost semi-local models, achieving significant error reduction compared to standard energy-density training.
\end{itemize}

\section*{Weaknesses}
While the method is technically interesting, I have reservations regarding its novelty and the scope of its impact, leading to a score of 4.

\begin{itemize}
    \item \textbf{Incremental Novelty:} The core idea of regularizing XC learning using potential-derived quantities is not entirely new. The paper cites related works like Kanungo et al. (2025) and Li et al. (2021) which also use potential or trajectory-based regularization. While the specific formulation using the Grassmannian Hessian is new, it feels like an incremental technical refinement rather than a paradigm shift. The connection to Deep Equilibrium Models (DEQs) is mentioned but effectively boils down to "matching the Jacobian," which is a known stabilization technique.

    \item \textbf{Reliance on Distillation:} The primary success shown is distilling B3LYP into a semi-local form. While useful for accelerating calculations, this does not address the harder problem of discovering functionals that outperform existing approximations (e.g., approaching CCSD(T) accuracy where B3LYP fails). The method relies on having access to high-quality gradients and Hessians from a "teacher" functional. It is unclear how this method scales to learning from ab initio data (like CCSD(T)), where obtaining the "true" XC potential and its Hessian is computationally prohibitive or ill-defined (requires inverse DFT).

    \item \textbf{Hyperparameter Sensitivity:} The method introduces several new hyperparameters ($\alpha_\nabla$, $\alpha_H$, and the adaptive stabilization schedule). The appendix suggests a complex tuning process. There is a concern that the method is somewhat fragile and requires extensive tuning of loss weights to balance the competing objectives of energy, density, and curvature matching.

    \item \textbf{Limited Chemical Space:} The evaluation is largely restricted to QM9 (small organic molecules). The "generalization" claims are mostly about basis set size or slightly larger molecules within the same chemical domain. It remains unproven whether this regularization helps in challenging regimes where ML-XC typically fails, such as transition metal complexes, bond breaking, or systems with strong correlation.

    \item \textbf{Complexity vs. Gain:} The training procedure is significantly more complex than standard training (requiring HVPs, specific initializations, and adaptive rejection). While the resulting inference is fast, the training overhead and implementation complexity are high. The improvement in energy MAE (e.g., in Table 4) is notable but not always drastic compared to the simpler "Energy + Density" baseline for some metrics.
\end{itemize}

---

> ### Author Rebuttal · Authors · 2026-03-31
>
> **Note to all reviewers:** We direct your attention to the *Important Remark* in our response to Reviewer Taqb.
>
> ## W4: Limited chemical space in evaluations
> Thanks for the feedback! We will add this paragraph to the limitations:
> >All experiments in this work are restricted to closed-shell organic molecules, a relatively narrow chemical domain. Whether the benefits of DI-Loss transfer to periodic solid-state systems remains an open question. Additionally, while Hessian supervision improves the quality of the learned functional, it introduces a non-negligible training cost overhead ($\sim 33$\%). However, this overhead is confined entirely to training. The resulting semi-local functional incurs no additional inference cost.
>
> ## W2 and Q1: Scalability to wavefunction data:
> DI-Loss is indeed most directly applicable to the distillation setting. We discuss this limitation and a concrete multi-stage training strategy in Section 6 (see "Limitations and Future Work").
>
> We also note that the current ML-XC state of the art trained on CCSD(T) data (Skala [1]) only outperforms hybrids and range-separated hybrids on the atomization energies it was explicitly trained on. On barrier heights, Skala's MAD of 4.05 kcal mol⁻¹ is worse than the B3LYP (3.60), ωB97M-V (1.84), and the double-hybrid ωB97M-2 (1.19) [2], suggesting that hybrid-level accuracy across chemical properties remains a non-trivial target even with wavefunction-level training data. Our framework is in principle directly applicable to ωB97M-V distillation, but we considered implementing VV10 nonlocal correlation in our differentiable SCF beyond the scope of this proof-of-concept. More broadly, distilling through progressively scarcer rungs ((range-separated) hybrids $\mathcal{O}(N^4)$, double hybrids $\mathcal{O}(N^5)$, followed by fine-tuning on CCSD $\mathcal{O}(N^6)$ or CCSD(T) $\mathcal{O}(N^7)$ energies) could offer a natural curriculum making best use of available data at each level, though this remains future work.
>
> ## Q2: Hyperparameter robustness
> We ablated $\alpha_\nabla$ and $\alpha_H$ independently, but all main-body results use the same weights per target functional (exception: EG-XC, where non-local features shift loss-term magnitudes). SCAN and B3LYP prefer different weights — unsurprisingly, since SCAN is nearly exactly representable at the semi-local level while B3LYP is not — but SCAN served primarily as a mechanistic study and is less sensitive overall. NNmGGA and Skala share the same weights without per-architecture tuning, though Skala may benefit from it.
>
> ## W1: Incremental novelty & Q3: Comparison to potential regularization
> We appreciate the reviewer's careful contextualization of our work and address each
> comparison in turn.
>
> Li et al. (2021) demonstrated that end-to-end training through the SCF solver has a regularizing effect. This is orthogonal to our contribution, as we also differentiate through the solver. DI-Loss is an additional training signal on top of end-to-end SCF training, not an alternative to it.
>
> Kanungo et al. (2025) supervise a density-weighted integral of $\Delta v_\text{xc}$, a single scalar moment, and do so in combination with a donated ground-state density to learn self-consistency without solver differentiation. This is a fundamentally different approach. In contrast, DI-Loss provides $O \times V$ independent directional constraints away from equilibrium (gradient) and stochastic curvature information at equilibrium (HVP). To further clarify the distinction, we tested a direct $V_\text{xc} \in \mathbb{R}^{B \times B}$ constraint as a drop-in replacement for our Grassmannian formulation in the B3LYP distillation setting. Direct $V_\text{xc}$ supervision degrades performance relative to the baseline across nearly all metrics:
>
> |$\alpha_{V_\text{xc}}$|$E_\text{tot}$|$E_\text{xc}$|$\Delta\varepsilon_\text{HL}$|$E_\rho$|$\mu_\rho$|$E_C$|
> |-|-|-|-|-|-|-|
> |(Ours)|1.57|1.66|49.35|0.87|0.0079|9.41|
> |0.0001|2.02|2.39|49.45|1.28|0.0617|343.08|
> |0.001|3.02|3.77|49.68|1.47|0.0321|78.42|
> |0.01|3.58|4.16|40.94|1.56|0.0271|28.28|
>
> Regarding the DEQ connection: to our knowledge, Bai et al. (2021) *constrain* $\lVert J_f \rVert_F$ to keep the spectral radius below 1 but do not *match* it to a target. We are unaware of DEQ work that supervises the Jacobian against a reference and would welcome a pointer if the reviewer has one. The key distinction is that constraining $\rho(J) < 1$ ensures convergence to *some* fixed point but encodes no physics; matching the orbital Hessian $H_{iajb}$ to a reference encodes $f_\text{xc}(\mathbf{r}, \mathbf{r}')$, which enters directly into the Casida equations for TDDFT excitation energies.
>
> ## Q4: Computational cost of training
> See answer to Taqb Q2
>
> [1] Luise et al., doi:10.48550/arXiv.2506.14665
>
> [2] Karton, doi:10.1021/acs.jpca.6c00424.

---

### Official Review · Reviewer_Taqb · 2026-03-12

**Soundness:** 3
**Presentation:** 3
**Significance:** 3
**Originality:** 3
**Overall Recommendation:** 5
**Confidence:** 3

**Summary:**

​This paper proposes a new loss function for learning exchange-correlation functionals. Building on conventional supervision on energies and densities, the method further incorporates first- and second-order derivative information. The experimental results show that, compared with training without derivative supervision, the proposed method can effectively reduce energy- and density-related errors. In addition, the authors demonstrate its practical potential, including improving SCF convergence efficiency and enhancing TDDFT excited-state predictions.

**Compliance With Llm Reviewing Policy:**

Affirmed.

**Final Justification:**

the authors has addressed my concerns and i will keep my positive score

**Key Questions For Authors:**

1. Could the authors provide comparisons with other regularization methods to more clearly demonstrate the unique value of derivative supervision?
1. Could the authors further quantify the additional cost of label generation and training computation introduced by the first- and second-order derivative supervision?

**Limitations:**

yes

**Strengths And Weaknesses:**

Strengths

1. The paper extends the training objective from matching only the final outputs to also matching derivative information, which is a reasonable idea with some degree of novelty.
2. The method is validated across multiple model architectures and evaluation metrics, making the empirical study fairly comprehensive.
3. Beyond improving fitting accuracy, the paper also shows benefits for iterative convergence efficiency and downstream task performance, which strengthens its practical value.

Weaknesses

1. The paper treats derivative supervision as a form of regularization, but currently lacks a clearer theoretical explanation, as well as direct comparisons with other regularization methods.
2. Incorporating first- and second-order derivative supervision requires additional label generation and loss computation, which introduces extra training cost, but this overhead is not analyzed in sufficient detail.
3. From the experimental results, first-order derivative supervision seems to contribute most of the performance gains, while the additional benefit of second-order supervision appears relatively limited and not always consistent.

---

> ### Author Rebuttal · Authors · 2026-03-31
>
> # Important Remark
> During early development with the Adam optimizer, we performed extensive ablations over density loss types and identified the mean-field density error $E_\rho$ as the strongest baseline. When we later adopted Muon, all models improved substantially, but we did not revisit the density-loss-type ablation. During rebuttal preparation, we repeated this ablation and found that $L_1$ density loss with Muon yields a baseline that nearly saturates the energy and density metrics of our small-scale OOD benchmark.
>
> |idx|$E_\mathrm{tot}$|$E_\mathrm{xc}$|$\Delta\varepsilon_\mathrm{HL}$|$\mu_\rho$|$E_C$|RIC|Warm RIC|
> |-|-:|-:|-:|-:|-:|-:|-:|
> |$E + \rho_{L_1}$|1.34|1.08|54.71|0.0107|6.82|1.10|0.47|
> |$+ \nabla$|1.27|1.38|57.22|0.0115|9.46|1.11|0.47|
> |$+ \nabla + H$|1.35|1.52|52.83|0.0100|7.14|1.09|0.46|
> |$+ H$|1.10|0.99|52.22|0.0093|5.50|1.09|0.45|
>
> This is a testament to the months of careful engineering that benefits all models equally:
> * optimizer selection (Muon)
> * improved model initialization
> * Training stabilizer
> * Lorentzian broadening of degenerate eigenvalues (necessary for cylindrically symmetric molecules. Previous works seem to have NaN skipped their updates)
> * Tikhonov regularization of Pulay equations (better backward-pass conditioning of DIIS)
> * ...
>
> We emphasize that this is not an off-the-shelf baseline; it is the product of the same infrastructure we developed here. Kanungo et al. [3] offer a lens for understanding this shift. They showed that model XC potentials carry $\mathcal{O}(10^{-1}\text{–}10^{0})$ relative errors even when densities agree to $\mathcal{O}(10^{-3}\text{–}10^{-2})$: the density is a weak probe of functional quality. However, when the density fit is tight enough, differentiating through the SCF solver implicitly constrains the first-order potential landscape near the ground state, because the solver trajectory itself encodes $v_\text{xc}$ information. This might explain why Muon + $L_1$ absorbs much of the benefit that explicit gradient supervision ($+\nabla$) provided under Adam: the optimizer now drives the density close enough that the backward pass through the solver already provides a sufficiently accurate first-order signal. What solver differentiation does *not* capture is the second-order Hessian, which must still be explicitly supervised. With this new baseline, we observe the following:
>
> - **TDDFT** (Fig. https://figshare.com/s/b2747cb1305f6008115f): Hessian supervision still significantly reduces excited-state MAE. Notably, gradient-only supervision ($+\nabla$) *degrades* TDDFT predictions, while the Hessian term drives the improvement. This is mechanistically expected: TDDFT excitation energies depend on the second functional derivative $\delta^2 E_\text{xc}/\delta\rho\,\delta\rho'$.
>
> - **Far OOD extrapolation** (QM40 [2]; Fig. https://figshare.com/s/61d13abfb6bdc1e5a931): On molecules with up to 40 heavy atoms (far beyond the 7-heavy-atom training distribution), DI-Loss ($+\nabla+H$) shows material improvement:
> |Model|$E_\text{tot}$|$E_\text{xc}$|$\Delta\varepsilon_\text{HL}$|$E_\text{MF}$|$\mu_\rho$|$E_C$|
> |-|-:|-:|-:|-:|-:|-:|
> |Baseline|4.61|3.45|51.82|1.51|0.0361|31.46|
> ||6.77|4.16|51.34|2.80|0.0770|53.52|
> ||5.07|3.47|51.43|1.74|0.0206|22.78|
> |DI-Loss|2.73|2.83|50.87|0.91|0.0226|26.75|
> ||3.46|3.41|50.92|0.68|0.0216|31.03|
> || 3.83 | 3.45 | 50.97 | 0.79 | 0.0219 | 32.45 |
> Each row in the table is an independent random seed
>
> - **SCF acceleration**: Warm-start RIC $\approx 0.50$ for both baseline and DI-Loss on **both** QM9 and QM40, confirming that both produce useful initial guesses for hybrid SCF.
>
> ## Q1: Regularization Techniques
> We appreciate this question. While DI-Loss does have a regularizing effect, characterizing it purely as regularization understates its role. Our pipeline already includes standard regularizers (KSR [2], weight decay). DI-Loss operates on a different level: it supervises the XC potential and XC kernel, i.e. physically meaningful quantities with direct downstream consequences. $V_{xc}$ governs SCF convergence, while $f_{xc}$ enters the Casida equations for TDDFT excitation energies. We will sharpen this distinction in the camera-ready version.
>
> ## Q2: Training Cost
> The derivative labels are computed on-the-fly during training. The additional cost is reflected in the per-train-step wall-clock times measured for the NNmGGA-B3LYP distillation on a single NVIDIA H100 GPU:
>
> |Loss|Time / step|
> |:-|:-|
> |Baseline ($E + \rho$)|~300 ms|
> |+ Gradient ($\nabla$)|~310 ms|
> |+ Hessian ($H$)|~400 ms|
>
> The gradient term adds negligible overhead (\~3%). The Hessian term incurs a more noticeable but modest increase (\~33%), which we consider acceptable given the consistent accuracy gains it provides.
>
> [1] Madushanka et al. doi: 10.1038/s41597-024-04206-y
>
> [2] Li et al. doi: 10.1103/PhysRevLett.126.036401
>
> [3] Kanungo et al. doi: 10.1021/acs.jpclett.1c03670.

---

> > ### Author Rebuttal · Reviewer_Taqb · 2026-04-03
> >
> > thank you for the response, i will keep the positive score.

---

### Official Review · Reviewer_e5NR · 2026-03-12

**Soundness:** 3
**Presentation:** 3
**Significance:** 2
**Originality:** 2
**Overall Recommendation:** 4
**Confidence:** 4

**Summary:**

This paper introduce a new data-driven DFA paradigm where one distill the $O(N^{4})$ hybrid XC into semi-local and non-local neural network XC functionals with $O(N^{3})$ scaling. The key contribution of the paper is to demonstrate empirically that training objective containing gradient and hessian of the XC functional with respect to the orbital coefficients performs better. In particular the derivatives is taken along orbital-rotation / Grassmannian direction in order to reduce cost. Besides improved energies, the experiments further demonstrated that densities, within-family basis transfer, SCF initialization, and TDDFT behavior is also improved with the gradient and hessian enhanced training objective.

**Compliance With Llm Reviewing Policy:**

Affirmed.

**Final Justification:**

The author has provided a detailed follow-up. The additional end-to-end wall-clock measurements directly address my main practical concern. In particular, the new results show that using the learned NNmGGA as a pre-run can produce a net speedup over direct B3LYP on the tested QM40 example, rather than merely reducing iteration count while hiding extra neural-network cost. While broader timing evidence would still strengthen the final paper, this is sufficient to resolve my original criticism that the acceleration claim lacked runtime support.

The clarification on basis-set transfer also resolves my earlier misunderstanding. I now understand that the OOD evaluation compares student and reference within the same test basis after training on another basis, so I withdraw my earlier statement that the result was expected simply because def2-SVP is a smaller basis. The added cross-family evaluation on 6-311+G(d,p) further strengthens the basis-transfer claim beyond within-family def2 transfer.

The training-overhead discussion is also helpful. The reported per-step overhead of the Hessian term, together with the wall-clock training curves, makes the practical cost of DI-Loss much clearer than in the original submission. I also appreciate the authors explicitly acknowledging the current empirical scope limitations, including the lack of solid-state evaluation.

I still view the empirical scope as somewhat narrow, and I would have preferred fuller multi-seed uncertainty reporting for the main hybrid-distillation results, but after the rebuttal, these read to me as limitations rather than blockers. Overall, the rebuttal materially improves my confidence in the paper, and I will adjust my score upward modestly.

**Key Questions For Authors:**

- Concerning the basis set generalization, shouldn't you be using a different basis family entirely (like the cc basis set) rather than using the smaller basis from the same family? It is expected that using a smaller basis will incur higher error.
- Are the main B3LYP distillation, SCF-initialization, and TDDFT results averaged over multiple random seeds, or are they single runs? Please report mean ± std (or confidence intervals) for Figures 3–5 and the selected Table 5 settings. This would materially change my confidence in the soundness of the main claims because the earlier SCAN tables do report uncertainty, while the main distillation section does not.
- What is the actual overhead of DI-Loss during training, especially for the Hessian term?
- Are the data for hybrids relatively more abundant than higher quality ab-initio data, such as CCSD?

**Limitations:**

Partially. The paper does include a useful limitations/future-work discussion. But I would still encourage the authors to explicitly add the limitation of the narrow empirical scope, as all experiments are done with closed-shell organics from QM9.

**Strengths And Weaknesses:**

Strength:
- The paper proposed a straightforward improvement for data-driven DFA that is easy to incorporate, where reference gradient and Hessians are available.
- The choice of distilling hybrids instead of the many-body method is sensible.
- The experimental result supports the claim that the loss enhanced with the gradient and Hessian term improves the result.
- I also liked that the paper does not rely on a single metric: page 5 Figure 1 and page 6 Figure 2 examine energies, density errors, HOMO-LUMO gaps, basis-set transfer, and even energy-density misallocation, which is much more convincing than an energy-only story.
- I appreciate the insight where the paper draws a connection between the SCF cycle and the DEQ model.
- The paper is easy to follow and well structured.

Weakness:
- For me, the main point of using ML-XC is to accelerate SCF runtime compared to running the existing XC functional. For the setup of this paper, the author needs to demonstrate that the ML-XC does indeed achieve speed up compared to running normal DFT with existing hybrids like B3LYP. However, the author did not report the inference time for running the SCF cycle with the learned XC. In Figure 4, the author reports reduced iteration counts when using the ML-XC density as initialization as compared to the SCAN density, but the runtime of ML-XC SCF pre-run should be reported here. It could be the case that the pre-run has a high cost due to the use of a neural network. Furthermore, the XC grid setting should be reported, as the grid level greatly impacts the SCF runtime. Another thing I would like to mention is that for molecular systems, using hybrid XCs does not incur prohibitive cost as compared to solid-state systems. The claim would be much stronger if the experiment were done with solid-state systems.
- The method is only tested on the narrow molecular dataset QM9. Doing a larger-scale experiment, and including solid state benchmark would greatly enhance the claim made by this paper.
- The originality is moderate. Enhancing loss with gradient and Hessian is already proposed in the PINN community as Sobolev training. The fact that the orbital coefficient lies on the Grassmannian, and the analytical formula for the gradient and Hessian with respect to the orbital coefficient, are textbook knowledge. The randomly sampled HVP is also a commonly used trick (for example, in diffusion model training for score matching).
- The main distillation results (Table 5 and Figures 3–5) are presented without uncertainty bars, unlike Tables 3–4, so it is hard to tell how stable the B3LYP-distillation and TDDFT gains are across random seeds.

---

> ### Author Rebuttal · Authors · 2026-03-31
>
> **Note to all reviewers:** We direct your attention to the *Important Remark* in our response to Reviewer Taqb.
>
> ## W1: SCF-Cycle Inference Time
> This is an important point indeed. We provide **single-cycle SCF walltime** benchmarks on QM9 molecules (1 cycle, averaged over 10 repetitions, 1× NVIDIA H200 GPU) comparing B3LYP against our NNmGGA across basis sets and system sizes. All timings are from our unoptimized research codebase.
>
> |Basis|#Heavy|#Atoms|#Basis|#Grid|B3LYP (ms)|NNmGGA (ms)|Speedup|
> |:-|-:|-:|-:|-:|:-:|:-:|-:|
> |def2-SVP|10|30|240|101240|19.0 $\pm$ 0.3|17.3 $\pm$ 0.1|1.10x|
> ||19|42|381|155216|94.3 $\pm$ 1.5|53.4 $\pm$ 0.1|1.77x|
> ||30|58|560|224376|294.1 $\pm$ 11.0|113.5 $\pm$ 1.6|2.59x|
> ||37|67|668|265340|596.4 $\pm$ 11.5|186.9 $\pm$ 2.5|3.19x|
> |def2-TZVPD|10|30|550|101240|150.7 $\pm$ 5.8|56.3 $\pm$ 0.7|2.67x|
> ||19|42|916|155216|1004.2 $\pm$ 11.0|182.0 $\pm$ 3.6|5.52x|
>
> Speedups increase consistently with system size, reaching over 5× at the def2-TZVPD level for 19 heavy atoms, reflecting the favourable $\mathcal{O}(N^3)$ vs. $\mathcal{O}(N^4)$ scaling of NNmGGA relative to B3LYP. At small system sizes the neural network evaluation introduces a modest overhead, but this is quickly amortised as the exact-exchange cost of the hybrid dominates.
>
> **We use PySCF grid level 1 throughout to reduce peak memory consumption**, as our current implementation of the Coulomb term requires the density-fitted ERI tensor to be kept in GPU memory rather than being rematerialized on the fly as in production-ready (inference-only) DFT codes.
>
> For a fully optimised, production-ready integration of the Skala architecture, we refer the reviewer to Section 5 and Figure 9 of Luise et al. [3], which reports GPU-accelerated runtimes matching semi-local meta-GGA (r2SCAN) cost, more than 10× below hybrid cost, with a CPU-based PySCF implementation showing a ∼3–4× prefactor over r2SCAN consistent with our findings above.
>
> ## Q1: Basis set Generalization
> > It is expected that using a smaller basis will incur higher error.
>
> We would like to clarify that the errors reported are always comparing the distillate and reference functional in the **same** basis set. In the specific case of the OOD (def2-SVP) test, we trained the functionals on the SCAN/def2-TZVPD level of theory and then evaluated them at the SCAN/def2-SVP level of theory (e.g. $\text{MAE}(E_\text{tot}) = |E^\text{(SCAN/def2-SVP)}_\text{tot} - E^\text{(ML-XC/def2-SVP)}_\text{tot}|)$
>
> ## Q2: Run-by-Run Variance of B3LYP Experiments
> We completly agree, and to the best of our knowledge prior work in ML-XC has neither compared against models not proposed by the same authors nor provided any run-by-run variance reports. We are therefore glad to be held to this higher standard. We did report variance wherever our compute budget permitted. The mGGA distillation runs (SCAN-family, reported with mean ± std) fit on NVIDIA A100 GPUs, of which we have ample availability. The hybrid functional distillation runs (B3LYP, Figures 3–5) require H100 GPUs, of which we have significantly fewer, and additionally use larger models trained on a larger subset of QM9 (molecules with up to 7 heavy atoms rather than up to 5), making individual runs substantially longer. Replicating these runs across multiple seeds within the rebuttal period is therefore not feasible.
>
> We want to emphasize, however, that the reported results are not single cherry-picked runs. Every individual run in the manuscript uses a distinct, hardware-entropy-sampled random seed. More importantly, to guard against seed-induced cherry-picking across the architecturally similar mGGA models in Figure 3, we fixed identical hyperparameters across architectures ($\alpha_\rho = 0.01$, $\alpha_H = 10^{-4}$) rather than tuning per model. The breadth of Table 5 is specifically intended to provide the variance signal the reviewer is rightly asking for: consistent trends are visible across all reported settings, including the ablation of the Hessian loss term without the gradient term, which appears particularly beneficial for energetic metrics of the Skala architecture and for the RIC between distillate and reference functional.
>
> ## Q3: DI-Loss overhead
> See awnser to EnkS Q4
>
> ## Q4: Data Availability Hybrid-DFT vs CCSD(T)
> Yes, the gap is large. For example: ANI-1x contains ~5M DFT conformations vs. ~500k CCSD(T)*/CBS points in its coupled-cluster counterpart, restricted to smaller molecules [1]. The $\mathcal{O}(N^4)$ vs. $\mathcal{O}(N^7)$ scaling gap makes large coupled-cluster datasets infeasible for all but the smallest systems [2], which is a key motivation for our distillation approach.
>
> Also see awnser (W2 & Q1) to EnkS
>
> ## Limitations: Empirical Scope
> see awnser W4 to EnkS
>
> [1] Smith et al., doi:10.1038/s41597-020-0473-z
>
> [2] Kulichenko et al., doi:10.1021/acs.chemrev.4c00572
>
> [3] Luise et al., doi:10.48550/arXiv.2506.14665

---

> > ### Author Rebuttal · Reviewer_e5NR · 2026-04-02
> >
> > Thanks for your detailed rebuttal. My concern about NN inference, derivative loss overhead, and basis set generalization has been partially resolved.
> >
> > My remaining concerns are:
> > 1. The new timing results support a per-cycle speed advantage over B3LYP. However, the end-to-end acceleration story for the workflow in Figure 4 is still incomplete without the total wall-clock time for “NNmGGA pre-run + B3LYP refinement” versus direct B3LYP.
> > 2. Lack of solid-state experiments.
> >
> > Follow-up questions:
> >
> > 1. Can you quantify the training-time overhead of DI-Loss relative to the baseline loss, especially for the Hessian-vector term?
> > 2. The clarification on the within-family basis transfer is helpful. Do you have any evidence, even on a small pilot subset, regarding transfer across a different basis family (e.g., cc-pVXZ)?

---

> > > ### Author Response · Authors · 2026-04-02
> > >
> > > ## W1: End-to-End wall-clock timings
> > > We measured the end-to-end time on a large (35 heavy atoms) randomly selected molecule of QM40 at the B3LYP/def2-SVP level of theory.
> > > We have confirmed that all SCF calculations have converged to exactly the same energy (-1587.69540087 $E_h$).
> > > The table shows the net wall-time speedup for different workflows (top: baseline; bottom: starting from B3LYP fully converged ML-XC calculation).
> > >
> > > |Workflow|ML-XC cycles|B3LYP cycles|Total cyc|Wall time (s)|Speedup|
> > > |:-|-:|-:|-:|:-|-:|
> > > |Direct B3LYP|0|14|14|4.969 $\pm$ 0.007|1.00|
> > > |ML-XC + B3LYP|4|11|15|4.422 $\pm$ 0.008|1.12|
> > > ||5|11|16|4.553 $\pm$ 0.005|1.09|
> > > ||6|8|14|3.673 $\pm$ 0.008|**1.35**|
> > > ||7|8|15|3.784 $\pm$ 0.006|1.31|
> > > ||8|8|16|3.899 $\pm$ 0.005|1.27|
> > > ||9|8|17|4.012 $\pm$ 0.006|1.24|
> > > ||10|8|18|4.123 $\pm$ 0.006|1.21|
> > > ||11|8|19|4.244 $\pm$ 0.006|1.17|
> > > ||12|8|20|4.355 $\pm$ 0.001|1.14|
> > > ||13|8|21|4.472 $\pm$ 0.008|1.11|
> > > ||14|8|22|4.586 $\pm$ 0.005|1.08|
> > >
> > > For this particular molecule, the sweet spot is at 6 NNmGGA cycles, yielding a 1.35x speedup at only 14 total cycles (matching the pure B3LYP count). We suspect even larger speed-ups for larger molecules.
> > >
> > > ## W2: Lack of solid-state experiments
> > > Thank you for highlighting this direction. We acknowledge that solid-state evaluation is a meaningful gap in the current work and have incorporated this into the Limitations section, as described in our response to Reviewer EnkS. We would be grateful for any pointers to particularly interesting solid-state datasets, as our expertise is primarily in molecular quantum chemistry rather than condensed-matter physics.
> > >
> > > ## Q1: Training-Time Overhead
> > > We include training wall-clock curves (Figure https://figshare.com/s/6e38fdf9f01947cd9ad3) showing training progress and validation energy error as a function of wall time for each loss variant. All runs use the same early-stopping parameters, and the majority of training sessions finish around the same time. The per-step overhead of the Hessian term (~33%) is visible in the left panel as fewer completed steps at a given wall time, but the total training duration remains comparable. Note that the validation set is drawn from the same QM7 distribution as the training data. The baseline's lower in-distribution validation error does not carry over to out-of-distribution settings (see our response to Reviewer Taqb). Here, we simply use the validation energy error as a common, physically interpretable observable to track training progress over time.
> > >
> > > Note that these curves are from the new $L_1$ density baseline runs (all run on the same H100 node with one H100 card used per run) referenced in the **run-by-run variance comparison between baseline and DI-Loss on B3LYP, including far-OOD evaluation on QM40** (See our *Important Remark* in response to Reviewer Taqb).
> > >
> > > For reference, we also provided per-step timings in our initial response (reproduced below for convenience), showing ~33% overhead from the Hessian term on a single H100 GPU:
> > >
> > > |Loss|Time / step|
> > > |:-|:-|
> > > |Baseline ($E + \rho$)|~300 ms|
> > > |+ Gradient ($\nabla$)|~310 ms|
> > > |+ Hessian ($H$)|~400 ms|
> > >
> > > ## Q2: Generalization Across Basis-Set Families
> > > Thanks for pushing on this. Basis-set transferability is an important practical concern, and we gladly address it in more detail.
> > >
> > > We reran the OOD evaluation on the 6-311+G(d,p) basis set using the same checkpoints as in Figure 1 of the paper (Figure https://figshare.com/s/035eac880c34ac9f4b0d). This provides evidence for transfer across basis families (Pople vs. Karlsruhe def2).
> > >
> > > We note that basis set generalization may be more challenging when distilling hybrid functionals into semi-local ML-XC functionals than when training meta-GGAs against purely semi-local targets. The exact-exchange component of the hybrid is computed directly from the orbital expansion coefficients and is thus expected to depend more strongly on the choice of basis set. A semi-local student that absorbs this basis-set-dependent signal into its predictions may overfit to the specific basis set used during training. This is less of a concern when the training target itself depends only on grid-based quantities ($\rho$, $\nabla\rho$, $\tau$), which converge smoothly towards basis-set-independent values. We will highlight this in our limitations section.
> > >
> > > Regarding cc-pVXZ specifically, to avoid PySCF becoming the training bottleneck, we reimplemented the basis-set evaluation on the integration grid in JAX. This reimplementation currently only supports segmented contraction schemes (as used by Pople and def2 families). The Dunning correlation-consistent basis sets use general contraction, where multiple contracted functions share the same set of primitives with different contraction coefficients, which our code does not yet handle. Extending the implementation is straightforward, but was not completed within the rebuttal period.

---

### Official Review · Reviewer_GRcc · 2026-03-12

**Soundness:** 4
**Presentation:** 4
**Significance:** 3
**Originality:** 3
**Overall Recommendation:** 5
**Confidence:** 4

**Summary:**

This paper proposes DI-Loss, a composite loss that supervises first- (gradient) and second-order (Hessian) derivatives of the energy on the Grassmannian manifold of density matrices for training ML exchange-correlation functionals. Beyond matching the SCF fixed point, DI-Loss aligns the energy landscape around the ground state with the reference functional. Experiments demonstrate improvements in energy/density accuracy, basis set generalization, SCF acceleration via warm-start, and TDDFT predictions, validated across multiple ML-XC architectures in mGGA-to-mGGA and hybrid-to-semilocal distillation settings.

**Compliance With Llm Reviewing Policy:**

Affirmed.

**Final Justification:**

While further large-scale empirical validation (e.g., SCF accelration) could strengthen the work, the theoretical grounding and the clear performance gains on the Grassmann manifold represent a significant step forward for the field. The insights provided in this study are valuable to both the ML and computational chemistry communities. Given the technical rigor and the potential for downstream impact, I recommend this paper for Acceptance.

**Key Questions For Authors:**

**Questions**

- **Wall-clock time (training / inference):** What is the wall-clock overhead of DI-Loss relative to $E+ρ$ training? What hardware was used, and what is the total compute budget? For the SCF acceleration experiment (Figure 4), how does the total wall-clock time of "distilled functional SCF + B3LYP warm-start" compare against "B3LYP from MINAO" end-to-end?
- **Model scaling:** Why was the model size limited to ~1,200 parameters? Have you observed any scaling trends for DI-Loss's marginal benefit with increasing model capacity? It would be informative to know whether larger models reduce the need for derivative supervision or continue to benefit from it.
- **Training dynamics:** Can you provide training curves (loss vs. step) comparing $E+ρ$ vs. DI-Loss? Specifically, does derivative supervision accelerate convergence to the same solution, or shift the converged solution itself? Understanding whether DI-Loss acts primarily as a regularizer (constraining the solution space) or an optimization aid (improving the training landscape) would clarify the mechanism.
- **Reproducibility:** Will code and the differentiable SCF implementation be released? Details on the SCF framework used (custom, DQC, PySCFAD, etc.) and compute environment would aid reproducibility.
- **Minor typographical issues:** Eqs. 11 and 13 appear to contain errors.
    - Eq. 11: may be the first term may need to be E_tot or E_non-xc rather than E_xc, as the current form adds the XC gradient twice.
    - Eq. 13: the denominator appears to be missing $∂θ_jb$ for a second-order derivative.

**Limitations:**

Yes

**Strengths And Weaknesses:**

**Strengths**

- **Well-motivated and self-contained.** The paper provides an accessible introduction to KS-DFT and the Grassmannian structure of density matrices, making the problem approachable for the ML community. The SCF-as-DEQ framing effectively bridges quantum chemistry and implicit deep learning.
- **Comprehensive evaluation.** The experiments cover multiple architectures (NNmGGA, XC-diff, Skala-mGGA, EG-XC), diverse metrics (energy, density, HOMO-LUMO gap, dipole, SCF convergence, TDDFT), both ID and OOD basis sets, and size extrapolation. The loss weight sweep (Appendix M) and optimizer ablation (Appendix H) are thorough. Cross-architecture consistency strengthens the generality claim.
- **Methodological contribution.** The decomposition of supervision into energy (value), density (fixed-point location), gradient (descent direction), and Hessian (curvature) is intuitive and sound. The PINN-style approach to XC functional learning is a natural and timely research direction with potential to become standard practice.


**Weekness / concerns**

- **DFT-to-DFT distillation only.** All experiments use DFT-level targets (SCAN, B3LYP). Since DI-Loss requires the reference functional's derivatives (unavailable for wavefunction methods like CCSD(T)), the method cannot directly address the most impactful setting. The proposed multi-stage strategy (DI-Loss pretraining → CCSD(T) fine-tuning) is unvalidated. Even a perfect B3LYP distillation inherits its known limitations (dispersion, barrier heights).
- **No wall-clock time analysis.** For a paper centered on computational efficiency (O(N⁴)→O(N³)), the absence of timing data is a significant gap: training overhead of gradient/Hessian computation, per-step SCF cost, and end-to-end comparison of "distilled warm-start" vs. "B3LYP from scratch" are all missing. The asymptotic scaling advantage may not materialize for QM9-sized molecules.
- **Very small model scale.** Semi-local models use ~1,200 parameters (Table 6); EG-XC is also down-scaled. Whether DI-Loss benefits persist at practically relevant model sizes is unknown. Larger models may not require derivative supervision, making the scaling behavior of DI-Loss's marginal benefit a critical open question.

---

> ### Author Rebuttal · Authors · 2026-03-31
>
> **Note to all reviewers:** We direct your attention to the *Important Remark* in our response to Reviewer Taqb.
>
> ## Q1: Wall-Clock Timings
> Training overhead and hardware: See answer to Taqb Q2
>
> SCF warm-start: At the scale of QM9 with def2-SVP (the setting of Figure 4), the per-cycle benchmarks (See answer to e5NR W1) imply no wall-clock improvement over B3LYP (~1× speedup at 10 heavy atoms). The SCF warm-start experiment in Figure 4 was designed to demonstrate that the distilled density provides a qualitatively better initialization than MINAO, reducing iteration counts as a proof of concept for larger-scale applications. The meaningful wall-clock benefit of this usage pattern arises at larger system/basis sizes, where the per-cycle speedup compounds with the reduced iteration count (See answer to e5NR W1).
>
> ## Q2: Model Sizes and Scaling Trends
> For the mGGA distillation experiments, model sizes were matched to those of XC-diff for fair comparability. The small size also allows runs to fit on a single NVIDIA A100 GPU, enabling us to investigate larger basis set experiments and run-by-run variance. We simply overlooked reporting the B3LYP model sizes in the original submission, and have added them to the appendix hyperparameter table:
>
> |Model|Task|\# Params|Hidden Dim.|Depth|
> |:-|:-|-:|-:|-:|
> |NN-mGGA|SCAN|1243|23|4|
> |XC-diff|SCAN|1250|16|4|
> |Skala-mGGA|SCAN|1212|22|4|
> |NN-mGGA|B3LYP|4417|32|6|
> |Skala-mGGA|B3LYP|4513|32|6|
>
> We have not observed qualitative differences in the relative benefit of the DI-Loss terms across model sizes. Interestingly, the additional expressivity of the larger models appeared particularly impactful for B3LYP distillation, likely because capturing the non-local character of exact exchange places greater demands on model capacity than semi-local functional distillation. A systematic scaling study of DI-Loss benefit with model size is an interesting direction we leave for future work.
>
> **The EG-XC architecture** combines a learnable meta-GGA with a non-local equivariant GNN backbone. For B3LYP distillation we downscaled only the GNN backbone while **upscaling** the meta-GGA part, and switched from XC-diff to NNmGGA because XC-diff enforces exact constraints violated by the hybrid target. In early ablations, the additional GNN capacity of EG-XC primarily improved energetic metrics without yielding corresponding gains in density metrics, as can be seen by comparing the energy-only rows in Table 5. EG-XC was initialized from the corresponding NNmGGA checkpoint for each loss configuration:
>
> |Model|$\alpha_\nabla$|$\alpha_H$|$E_\mathrm{tot}$|$E_\mathrm{xc}$|$\Delta\varepsilon_\mathrm{HL}$|$\mu_\rho$|$E_C$|RIC|
> |:-|-:|-:|-:|-:|-:|-:|-:|-:|
> |NNmGGA|$0$|$0$|2.02|1.67|53.95|0.0302|179.28|1.14|
> |+ EG-XC|$0$|$0$|1.17|1.81|53.68|0.0292|179.16|1.14|
> |NNmGGA|$10^{-2}$|$0$|1.57|1.66|49.35|0.0079|9.41|1.14|
> |+ EG-XC|$10^{-5}$|$0$|0.45|0.73|50.13|0.0088|25.25|1.15|
> |NNmGGA|$10^{-2}$|$10^{-4}$|1.44|1.33|47.47|0.0075|7.66|1.11|
> |+ EG-XC|$10^{-5}$|$10^{-5}$|0.54|0.63|48.29|0.0103|5.95|**1.09**|
>
> EG-XC seems to primarily improve the energy while leaving density metrics and RIC essentially unchanged. Notably, EG-XC initialized from pretrained NNmGGA prefers smaller $\alpha_\nabla$, suggesting the pretrained head already captures much of the gradient-level information, and the GNN backbone benefits from a softer derivative signal. Adding the Hessian term (row 6) yields the best RIC (1.09 vs 1.15), suggesting that the additional non-local capacity of EG-XC aids the joint reconciliation of matching energies and curvature, a trade-off that the smaller NNmGGA must navigate with less expressive capacity.
>
> ## Q3 Training Dynamics
> https://figshare.com/s/fd71adf6a625c598b0a9
> DI-Loss shifts the converged solution, not merely the path to it. This is evident from the density-derived metrics ($E_\rho$, $\mu_\rho$, $\Delta\varepsilon_\text{HL}$) differing between baseline and DI-Loss at convergence. If derivative supervision only accelerated convergence to the same solution, these metrics would be identical. We will add training curves to the camera-ready appendix.
>
> ## Q4: Reproducibility and Code Release
> We build on the autodifferentiable SCF implementation of Gao et al. [1], which we extended and improved in efficiency. We will release both the code and trained model weights alongside this work. All experiments are run on single GPUs in a university compute cluster (NVIDIA A100, H100, and H200 cards). We also attached our anonymized repository in the supplementary material.
>
> [1] Gao, N.; Eberhard, E.; Günnemann, S. Learning Equivariant Non-Local Electron Density Functionals, 2024.

---

> > ### Author Rebuttal · Reviewer_GRcc · 2026-04-03
> >
> > I thank the authors for the thorough rebuttal. My concerns have been adequately addressed. The wall clock analysis, model scaling discussion with the EG-XC ablation table, and training dynamics evidence showing that DI-Loss shifts the converged solution rather than merely accelerating convergence are all informative. I would recommend that the authors ensure proper citation and acknowledgment of Gao et al. [1] in the main text, given that the SCF implementation builds directly on their work. I believe this is a solid contribution ready to be presented to the community. I maintain my current score.

---

### Decision · Program_Chairs · 2026-04-30

**Decision:**

Accept (regular)

**Comment:**

The paper presents a derivative-informed XC loss (DI-Loss) for learning exchange-correlation functionals, extending supervision beyond energies to also include density, gradient, and Hessian information on the Grassmannian manifold of density matrices. The key motivation is that DI-Loss encourages the SCF dynamics to align with the target functional, rather than merely matching the final self-consistent fixed point. By distilling hybrid $O(N^4)$-scaling functionals into $O(N^3)$-scaling ML-XC functionals, the method effectively reduces both energy and density errors. The authors also demonstrate promising practical benefits, including improved SCF convergence and better TDDFT excited-state predictions.

All reviewers and I appreciate the technical contribution of incorporating gradient and Hessian information on the Grassmannian manifold into DFA learning. Reviewers also found the empirical study relatively comprehensive, particularly in terms of architectural coverage, evaluation metrics, and ablations. The rebuttal was helpful in addressing several important practical questions, including wall-clock cost, the training overhead of derivative terms, and basis-family transfer.

Reviewers also noted several limitations. First, the study remains empirically narrow, focusing mainly on small closed-shell organic molecules (Reviewers GRcc, e5NR, EnkS), with no validation on solid-state systems (e5NR), transition-metal chemistry (EnkS), or other regimes where ML-XC methods often struggle. Second, the main setting is still DFT-to-DFT distillation (GRcc, EnkS), so it remains unclear whether the method can extend toward wavefunction-level supervision. Third, Reviewer e5NR viewed the technical novelty as moderate rather than fundamental, since derivative-based training has precedents in related areas and some of the mathematical ingredients are already known.

Overall, these issues are minor relative to the strength of the proposed training strategy. I nevertheless encourage the authors to clarify these limitations more explicitly in the final version.